# Aberrant GlyRS-HDAC6 interaction linked to axonal transport deficits in Charcot-Marie-Tooth neuropathy

Zhongying Mo[1], Xiaobei Zhao[2], Huaqing Liu[1], Qinghua Hu[1], Xu-Qiao Chen[2], Jessica Pham[1], Na Wei[1], Ze Liu[1], Jiadong Zhou[1], Robert W. Burgess[3], Samuel L. Pfaff[4], C. Thomas Caskey[5], Chengbiao Wu [2,6], Ge Bai[1,4] & Xiang-Lei Yang[1]

Dominant mutations in glycyl-tRNA synthetase (GlyRS) cause a subtype of Charcot-Marie-Tooth neuropathy (CMT2D). Although previous studies have shown that GlyRS mutants aberrantly interact with Nrp1, giving insight into the disease's specific effects on motor neurons, these cannot explain length-dependent axonal degeneration. Here, we report that GlyRS mutants interact aberrantly with HDAC6 and stimulate its deacetylase activity on α-tubulin. A decrease in α-tubulin acetylation and deficits in axonal transport are observed in mice peripheral nerves prior to disease onset. An HDAC6 inhibitor used to restore α-tubulin acetylation rescues axonal transport deficits and improves motor functions of CMT2D mice. These results link the aberrant GlyRS-HDAC6 interaction to CMT2D pathology and suggest HDAC6 as an effective therapeutic target. Moreover, the HDAC6 interaction differs from Nrp1 interaction among GlyRS mutants and correlates with divergent clinical presentations, indicating the existence of multiple and different mechanisms in CMT2D.

[1] Department of Molecular Medicine, The Scripps Research Institute, La Jolla, CA 92037, USA. [2] Department of Neurosciences, University of California at San Diego, La Jolla, CA 92093, USA. [3] The Jackson Laboratory, Bar Harbor, ME 04609, USA. [4] Howard Hughes Medical Institute and Gene Expression Laboratory, The Salk Institute for Biological Studies, La Jolla, CA 92037, USA. [5] Baylor College of Medicine, Houston, TX 77030, USA. [6] Veterans Affairs San Diego Healthcare System, San Diego 92161 CA, USA. Correspondence and requests for materials should be addressed to X.-L.Y. (email: xlyang@scripps.edu)

Charcot-Marie-Tooth (CMT) disease is a group of genetically distinct disorders of the peripheral nervous system, with clinical presentations characterized by progressive muscle weakness, atrophy, and sensory loss in body extremities[1–3]. Collectively, the disease affects one in 2500 people worldwide, making it the most common inherited neuromuscular disorder[2]; however, no treatment is available for CMT patients. Based on the predominant pathological features, CMT is divided into two major types—type 1 where abnormalities occur in the myelin sheath surrounding peripheral axons (CMT1) and type 2, where the damage is within the axon itself (CMT2), though intermediate forms also exist[4]. CMT mainly affects long peripheral nerves, indicating a length-dependent axonal degeneration.

Aminoacyl-tRNA synthetases are the largest gene/protein family implicated in CMT[3]. Glycyl-tRNA synthetase (GlyRS or GARS) in particular was the first member identified, with over one dozen dominant mutations associated with CMT type 2D (CMT2D)[5–9]. The fundamental activity of this enzyme family in catalyzing the first reaction in protein synthesis in all cells[10, 11] contrasts with the extreme tissue specificity of the disease. Human GlyRS is composed of three domains: a metazoan-specific helix-turn-helix WHEP domain; the evolutionarily conserved catalytic domain; and anticodon-binding domain (Fig. 1a). CMT2D mutations are found in all three domains of GlyRS (Fig. 1a), and defective aminoacylation function is not shared among all CMT2D-associated mutants (GlyRS$^{CMT2D}$)[12, 13]. Genetic deletion of one *Gars* allele in mice to reduce GlyRS expression to 50% level does not yield any phenotype[14]; transgenic overexpression of wild-type (WT) GlyRS cannot rescue phenotypes in mouse and *Drosophila* models of CMT2D[15, 16]. These results indicate that CMT2D is not caused by a simple loss of WT protein function, and instead arises from abnormal activities of mutant GlyRS$^{CMT2D}$.

We hypothesized that the abnormal activity of GlyRS$^{CMT2D}$ starts at the level of protein structure, and found that different mutations associated with CMT2D cause a shared conformational opening effect in GlyRS that exposes new protein surfaces to solution[17, 18]. The neomorphic conformational opening correlates with aberrant interactions made by GlyRS$^{CMT2D}$ to Nrp1 and Trk receptors (through their extracellular domains) to explain the motor neuron and sensory neuron selectivity, respectively, of CMT2D[17, 19]. However, the neomorphic structural opening in principle would also enable GlyRS$^{CMT2D}$ to gain aberrant interactions with intracellular targets. Furthermore, an aberrant GlyRS-Nrp1 interaction cannot explain why peripheral nerves are selectively affected in CMT2D. These outstanding questions prompted us to look for aberrant interactions of GlyRS$^{CMT2D}$ that could impose a specific challenge to long peripheral nerves.

Deficits in axonal transport are a common theme in many neurodegenerative diseases[20, 21]. Long nerves are particularly vulnerable to axonal transport deficits, as they require extensive trafficking of structural elements and signaling molecules between cell body and nerve endings. In fact, many CMT-associated genes are directly or indirectly involved in axonal transport[22–24]. The cytoskeleton, especially microtubules, provide the tracks along which long distance axonal transport occurs[21]. A major component of the microtubule is α-tubulin, which undergoes post-translational modifications that regulate microtubule dynamics and functions[25]. In particular, acetylation of α-tubulin at Lys40 facilitates axonal transport by promoting binding of the motor proteins kinesin and dynein to the microtubules[26–28]. The removal of the modification is catalyzed by histone deacetylase 6 (HDAC6)[29, 30], whose inhibition has been shown to increase α-tubulin acetylation[31], rescue axonal transport defects[32], and provide benefits in animal models of neurodegenerative diseases, including a subtype of CMT2

(CMT2F)[23]. However, in most cases it is not clear whether the axonal transport defect is causatively linked to the neurodegenerative diseases, or merely a secondary consequence of other pathogenic factors.

In this study, we identify HDAC6 as an intracellular molecule that aberrantly interacts with GlyRS$^{CMT2D}$. The aberrant interaction promotes the deacetylase activity of HDAC6 and impairs α-tubulin acetylation. Peripheral nerve axonal transport defects are detected in CMT2D mice before the onset of neurodegeneration, suggesting that the deficits are not due to secondary effects. By using an HDAC6 inhibitor, we are able to rescue the axonal transport defect and improve motor functions in CMT2D mice. These results indicate that HDAC6 is an effective therapeutic target for CMT2D and that HDAC6 hyperactivation contributes to CMT2D as a pathogenic factor. Furthermore, the GlyRS$^{CMT2D}$-HDAC6 and GlyRS$^{CMT2D}$-Nrp1 interaction patterns differ from one another, in a way that correlates with divergent clinical presentations among various mutations, suggesting that multiple and different mechanisms exist in CMT2D.

## Results

**GlyRS$^{CMT2D}$ make aberrant interaction to HDAC6.** We searched in databases for potential interaction partners of GlyRS involved in axonal transport and found HDAC6 as a candidate[33]. To investigate the potential interaction, and to explore the effect of CMT2D mutations, we performed immunoprecipitations using neural tissues from WT (*Gars$^{+/+}$*) and P234KY-mutated (*Gars$^{P234KY/+}$*, numbered after the human protein, omitting the mitochondrial targeting sequence, also reported as P278KY) CMT2D mouse littermates. HDAC6 interaction is detected in CMT2D mice, but not in the WT control animals (Fig. 1b), indicating that only GlyRS$^{P234KY}$, but not GlyRS$^{WT}$, can interact with HDAC6 in vivo.

To understand whether the effect of P234KY is shared by other CMT2D-associated mutations, we transfected the mouse motor neuron cell line NSC-34 with V5-tagged GlyRS$^{CMT2D}$ constructs of nine different human mutations along with the C157R (also reported as C201R) and P234KY mutations found in CMT2D mouse models[14, 34]. Remarkably, all mutants exhibit aberrant interaction with HDAC6 (Fig. 1c). However, the strength of the interaction differs substantially among the mutants. In particular, the HDAC6 interaction with GlyRS$^{P234KY}$ is stronger than with GlyRS$^{C157R}$ (Fig. 1c), correlating with the severity of the phenotype of the two mouse models[14, 34]. Among the human mutations, the two anticodon-binding domain mutations S581L and G598A induce much stronger aberrant HDAC6 interaction than the other mutations (Fig. 1c). It is worth noting that all CMT patients carrying the G598A mutation exhibit extremely severe clinical symptoms and an infantile onset in contrast to the usual adolescence onset for CMT2D patients[35, 36]. However, the pathogenicity of the S581L mutation is unclear. Although the S581L mutation has been recurrently found in three unrelated CMT families[35, 37], it is also found at a low frequency (http://exac.broadinstitute.org/) in the general population[37, 38].

To clarify whether the HDAC6 interaction is specific for GlyRS$^{CMT2D}$, we further tested five missense mutations identified in the general population with similar frequency as that of the S581L mutation[38]. Interestingly, none of the five mutations induces the HDAC6 interaction (Supplementary Fig. 1), providing a distinction between S581L and the other variants identified in the general population. Also, consistent with our results from NSC-34 cells, the aberrant GlyRS-HDAC6 interaction is clearly detected in peripheral blood mononuclear cells (PBMCs) of a CMT patient carrying the S581L mutation, but not of a healthy volunteer (Fig. 1d).

**GlyRS$^{CMT2D}$ enhance HDAC6 deacetylase activity on α-tubulin**. To gain insight into the effect of the aberrant interaction, we mapped out the binding site on HDAC6 using GlyRS$^{P234KY}$. Human HDAC6 is a 1215-amino-acid protein with two catalytic domains, which are followed by a SE14 domain and a ubiquitin-binding domain (BUZ) that are responsible for cytosol retention and aggresome recruitment, respectively[39] (Supplementary Fig. 2a). Although the active site of HDAC6 resides in the second catalytic domain[40], spatial arrangement between the two catalytic domains is important, as alterations in

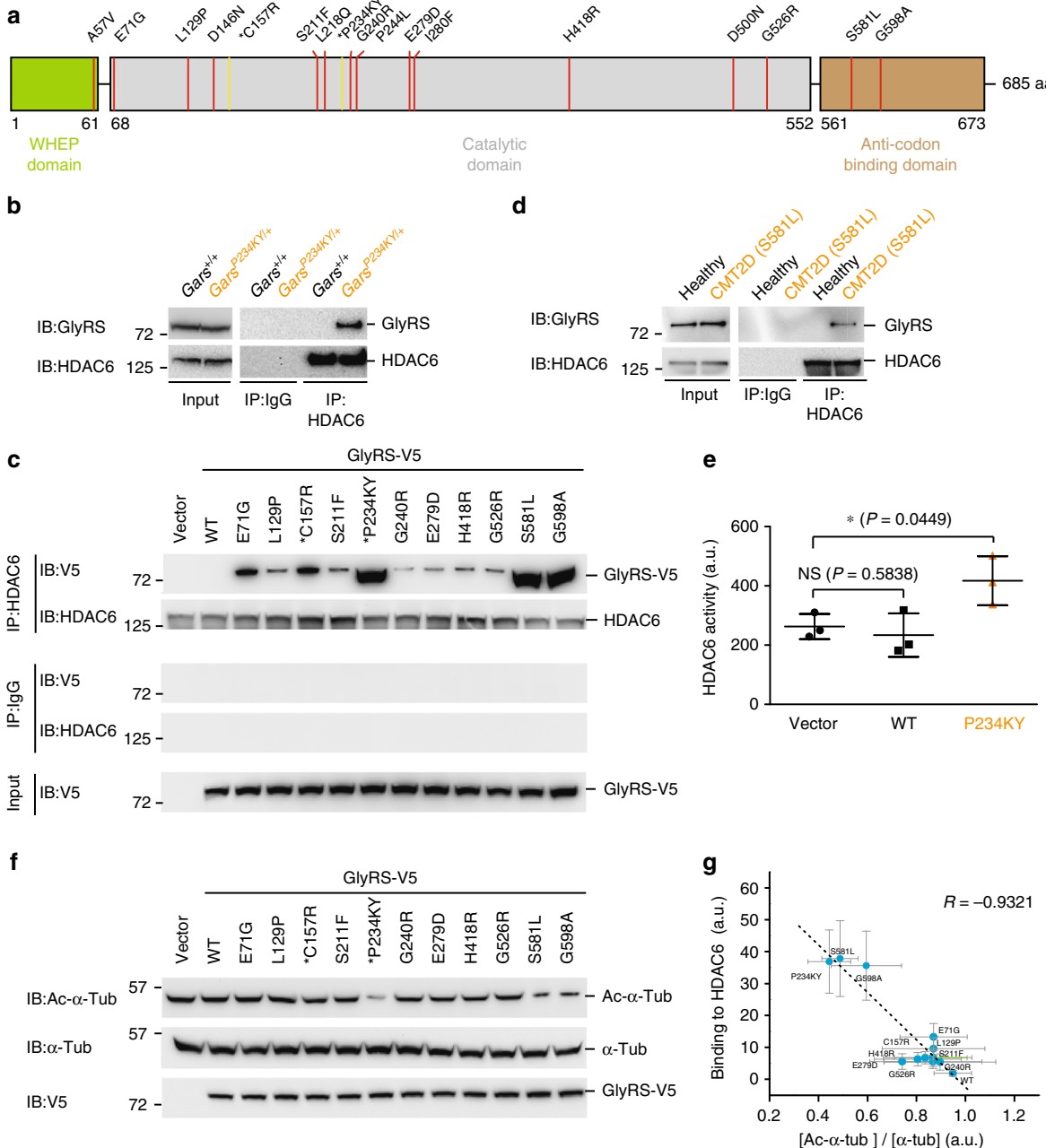

**Fig. 1** GlyRS$^{CMT2D}$ mutants bind to HDAC6 and enhance its deacetylation activity on α-tubulin. **a** CMT2D-associated mutations mapped on the three domains of human GlyRS. Two mutations identified in mice are labeled according to their residue numbers in the human protein and with asterisks. **b** Co-immunoprecipitation showing strong GlyRS-HDAC6 interaction in brain tissue of CMT (Gars$^{P234KY/+}$) mice but not WT (Gars$^{+/+}$) littermates (postnatal day 7). **c** Co-immunoprecipitation showing that GlyRS$^{CMT2D}$ proteins (C-terminal V5-tagged) but not GlyRS$^{WT}$ can bind to HDAC6 (endogenous) in transfected NSC-34 cells. **d** Co-immunoprecipitation showing GlyRS-HDAC6 interaction in peripheral blood mononuclear cells of a CMT patient carrying the GlyRS S581L mutation, but not of a healthy donor. **e** Overexpression of GlyRS$^{P234KY}$, but not GlyRS$^{WT}$, enhances HDAC6 deacetylase activity in HEK293 cell. Statistical analysis was done with two-tailed unpaired Student's t-test. Data are presented as means ± s.d. (n = 3 biological replicates per group). **f** Western blot analysis detecting the level of α-tubulin acetylation in NSC-34 cells transfected with various GlyRS (C-terminal V5-tagged) constructs. **g** Correlation analysis showing the relationship between the strength of an aberrant GlyRS$^{CMT2D}$-HDAC6 interaction and the acetylation level of α-tubulin in NSC-34 cells expressing various GlyRS proteins. The levels of α-tubulin, acetylated α-tubulin, and HDAC6-bound GlyRS were quantified with ImageJ and normalized against values of the vector control group. Data are presented as means ± s.d. (n = 3 biological replicates per group)

the linker region severely affect the deacetylase activity[41]. As shown in Supplementary Fig. 2b, removal of the BUZ domain or of both BUZ and SE14 domains does not seem to weaken the GlyRS$^{P234KY}$-HDAC6 interaction, and each of the two catalytic domains of HDAC6 alone can interact with the mutant GlyRS.

The involvement of the HDAC6 catalytic domains for the interaction suggests that GlyRS$^{P234KY}$ may influence the deacetylase activity of HDAC6. Indeed, significant enhancement of HDAC6 activity was observed in HEK293 cells with ectopic expression of GlyRS$^{P234KY}$, but not GlyRS$^{WT}$ (Fig. 1e). To understand whether the activation of HDAC6 affects α-tubulin, we detected the level of α-tubulin acetylation in NSC-34 cells expressing various GlyRS$^{CMT2D}$ mutations (Fig. 1f). The three mutations that induce the strongest HDAC6 interactions (i.e., P234KY, S581L, and G598A; Fig. 1c and Supplementary Fig. 3) also show greatly reduced levels of α-tubulin acetylation (Fig. 1f). Overall, a strong inverse correlation ($R = -0.9321$) was found in between the strength of an aberrant GlyRS-HDAC6 interaction and the acetylation level of α-tubulin (Fig. 1c, f, g), supporting the conclusion that aberrant GlyRS-HDAC6 interaction promotes the deacetylase activity of HDAC6 and leads to a decrease in α-tubulin acetylation.

**Reduced α-tubulin acetylation in CMT2D mouse sciatic nerves**. Next, we compared the levels of acetylated α-tubulin in CMT2D and control mice at different ages. A significant decrease in acetylated α-tubulin level was found in postnatal day 7 (P7) (Fig. 2a, b) and postnatal day 12 (P12) sciatic nerves (Supplementary Fig. 4a, b) of Gars$^{P234KY/+}$ mice compared to that of Gars$^{+/+}$ mice. Both time points precede the onset of CMT phenotypes, which happens around postnatal day 15–20[14]. Interestingly, the decrease in acetylated α-tubulin is specific to sciatic nerve and is not found in spinal cord or brain samples (Fig. 2a, b and Supplementary Fig. 4a, b), consistent with the peripheral nerve-selective pathology of the disease.

To understand the apparent peripheral nerve-specific decrease in α-tubulin acetylation, we compared the protein levels of GlyRS and HDAC6 between Gars$^{P234KY/+}$ and Gars$^{+/+}$ mice and did not observe significant difference (Fig. 2a, c and Supplementary Fig. 4c, d). However, we found that the level of HDAC6 is significantly lower in sciatic nerve than in spinal cord and brain (Fig. 2a), which is consistent with the relatively high acetylation level of α-tubulin in sciatic nerve in Gars$^{+/+}$ mice (Fig. 2a, b). In contrast, the level of GlyRS in the three tissue types is more or less similar (Fig. 2a). The relatively high level of GlyRS to HDAC6 in sciatic nerve (Fig. 2c) might provide the explanation for the peripheral nerve-specific decrease in α-tubulin acetylation in CMT2D mice.

HDAC6 has other substrates beyond α-tubulin. Among them, cortactin and HSP90 are the most studied[42, 43]. Interestingly, no significant difference in the levels of the acetylated cortactin and HSP90 is observed in between WT and CMT2D mice in any of the three types of neural tissue (Fig. 2a).

**Defective axonal transport precedes disease onset**. Because the acetylation of α-tubulin promotes the recruitment of motor proteins (for both anterograde and retrograde transport) to the microtubules[26, 27], the significant decrease in acetylated α-tubulin level in the sciatic nerves of Gars$^{P234KY/+}$ mice suggests potential axonal transport defects. We chose pre-symptomatic P12 mice for investigation to ensure that any potential axonal transport defect is not due to secondary effects of axonal degeneration. Dorsal root ganglia (DRG) of Gars$^{+/+}$ and Gars$^{P234KY/+}$ mice from the same litter were plated in microfluidic chambers to allow specific monitoring of axonal transport

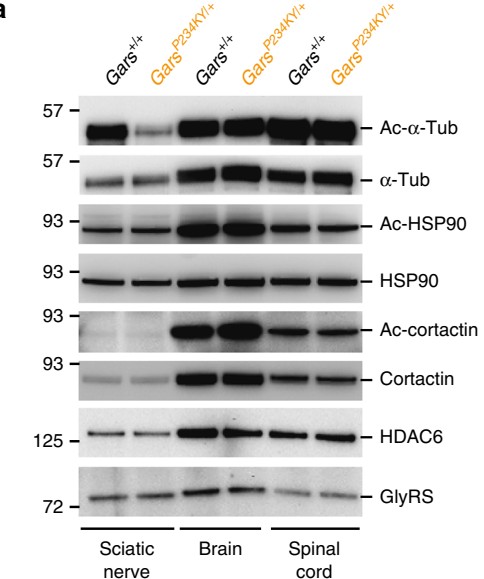

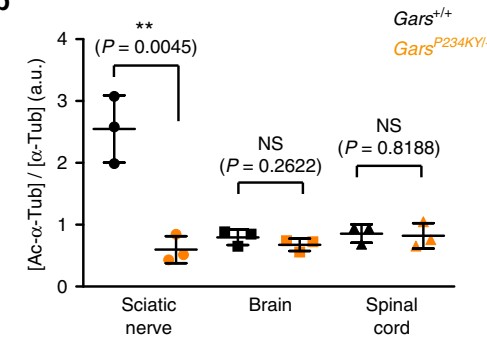

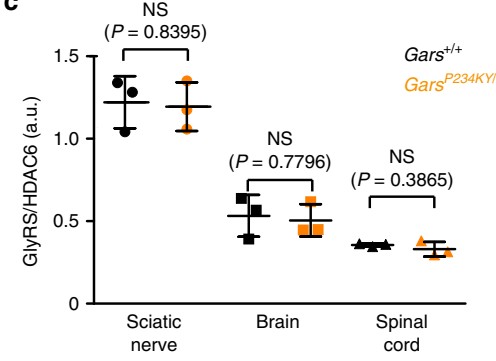

**Fig. 2** CMT2D mice exhibit decreased level of acetylated α-tubulin in sciatic nerves. **a** Western blot analysis showing decreased α-tubulin acetylation in sciatic nerves of CMT mice. No substantial change in cortactin and HSP90 acetylation was detected. Postnatal day 7 Gars$^{+/+}$ and Gars$^{P234KY/+}$ littermates were used for the analysis. Same amount of total protein (4 μg) was loaded in each lane. **b, c** Quantification of relative levels of acetylated α-tubulin (**b**) or GlyRS to HDAC6 (**c**) in three types of neural tissue. The protein levels were quantified with ImageJ. Statistical analysis was done with two-tailed unpaired Student's t-test. Data are presented as means ± s. d. ($n = 3$ mice per group)

(Fig. 3a, b and Supplementary Fig. 5). No difference in morphology and growth rate of the DRG axons was observed in between the Gars$^{+/+}$ and Gars$^{P234KY/+}$ cultures. Axonal transport was monitored by quantum dot-labeled nerve growth

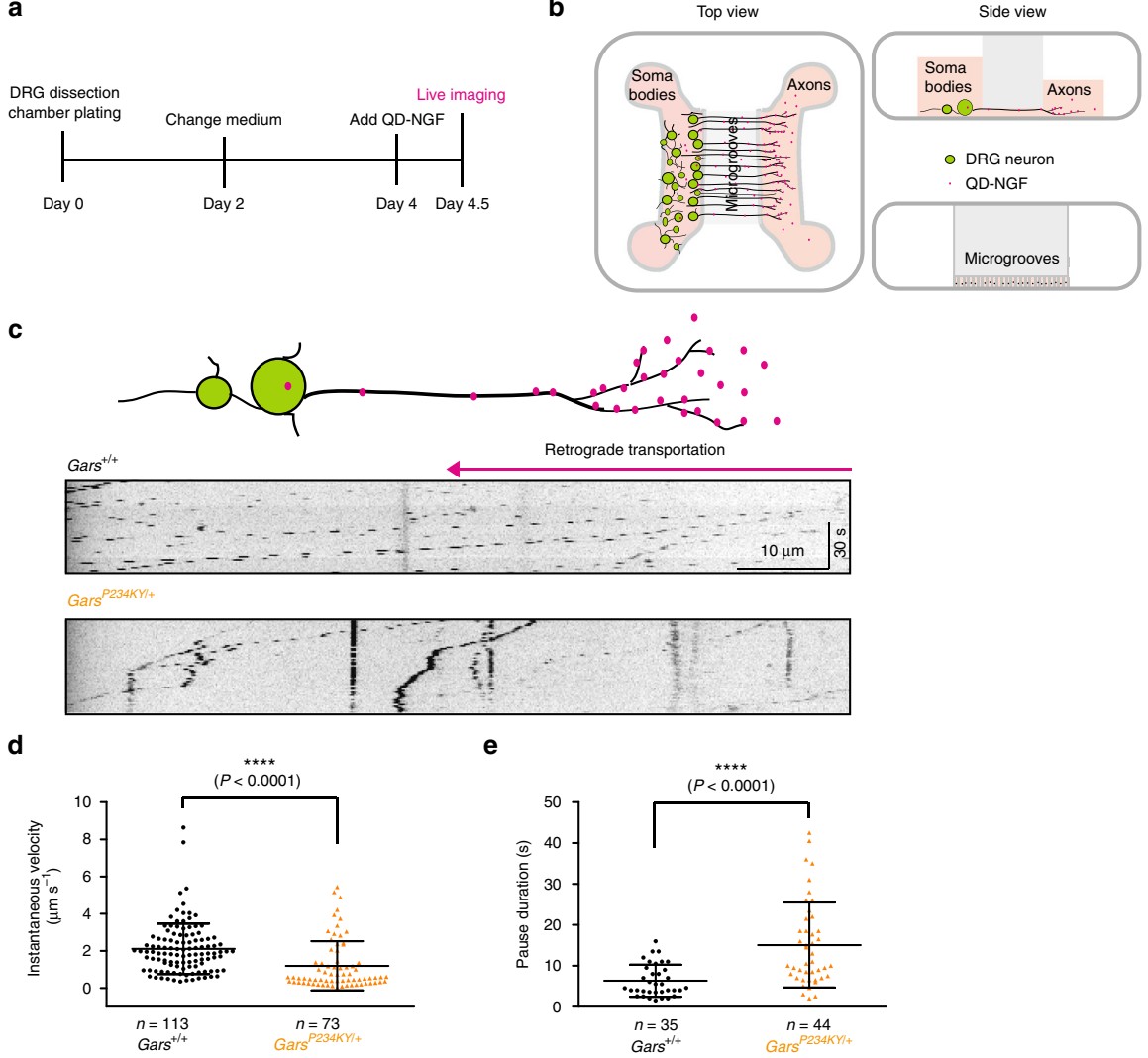

**Fig. 3** CMT2D mice exhibit axonal transport defect prior to disease onset. **a** Experimental design. **b** A schematic picture of the microfluidic chamber used to evaluate axonal transport of mouse DRGs. DRGs of postnatal day 12 mice ($Gars^{+/+}$ and $Gars^{P234KY/+}$ littermates) were plated in the cell body compartment. Axons grew across the microgrooves into the axon compartment after 3 days in culture. QD655-labeled NGF (QD-NGF) was added (0.2 nM final concentration) to the axon compartment for axonal transport live imaging. **c** Representative kymographs of QD-NGF transport in $Gars^{+/+}$ and $Gars^{P234KY/+}$ mice-derived DRG axons. **d**, **e** Instantaneous velocities and pause durations of QD-NGF transport in DRG axons from $Gars^{+/+}$ and $Gars^{P234KY/+}$ littermates. DRGs were dissected and pooled from three mice for each genotype. $n$ represents the number of QD-NGF-bearing endosomes measured for movements. Statistical analysis was done with two-tailed unpaired Student's $t$-test. Data are presented as means ± s.d.

factor (QD-NGF) added to the distal side of the DRG neurons (Fig. 3b, c). Retrograde axonal transport of NGF is critical for the survival and maintenance of peripheral neurons[44]. We found that the NGF transport in the axons of $Gars^{+/+}$ was characterized by rapid movements with an instantaneous velocity of $2.11 \pm 0.13 \, \mu m \, s^{-1}$ (mean ± SEM; Fig. 3c, d and Supplementary Movie 1). Almost all movements of QD-NGF were retrograde, but short-distance anterograde movements were occasionally observed. In contrast, the transport of NGF in the axons of $Gars^{P234KY/+}$ was significantly slower and interrupted by frequent pauses with an instantaneous velocity of $1.20 \pm 0.15 \, \mu m \, s^{-1}$ (Fig. 3c, d and Supplementary Movie 2). The pause duration showed a significant increase for the $Gars^{P234KY/+}$ ($15.08 \pm 1.57 \, s$) compared to that of $Gars^{+/+}$ ($6.37 \pm 0.66 \, s$) mice (Fig. 3e). Furthermore, there were substantial increases for the stationary (6.3% to 33.9%) and the anterograde transport (4.72% to 8.66%) events from $Gars^{+/+}$ to $Gars^{P234KY/+}$ mice. Therefore, $Gars^{P234KY/+}$ mice

exhibit significant axonal transport deficits in peripheral neurons prior to the onset of CMT phenotypes.

**HDAC6 inhibitor rescues the axonal transport defect**. To evaluate whether the axonal transport deficits are linked to HDAC6 overactivation, we used an HDAC6 inhibitor, tubastatin A (Tub A)[45]. We added Tub A ($2 \, \mu M$), or solvent control (0.02% dimethylsulfoxide (DMSO)) into the cultured DRG neurons from P12 $Gars^{+/+}$ and $Gars^{P234KY/+}$ mice to evaluate their effects on axonal transport (Fig. 4a). Tub A treatment does not affect QD-NGF transport in the axons of $Gars^{+/+}$ DRG neurons; instantaneous velocities are $2.87 \pm 0.26 \, \mu m \, s^{-1}$ and $2.92 \pm 0.26 \, \mu m \, s^{-1}$ under DMSO and Tub A treatment, respectively (Fig. 4b, c and Supplementary Movies 3 and 4). In contrast, Tub A treatment almost fully restored the retrograde axonal transport of QD-NGF in the $Gars^{P234KY/+}$ DRG, with an instantaneous velocity of $2.67 \pm 0.18 \, \mu m \, s^{-1}$ compared to $1.31 \pm 0.14 \, \mu m \, s^{-1}$ for the DMSO group (Fig. 4b, c and Supplementary Movies 5 and 6).

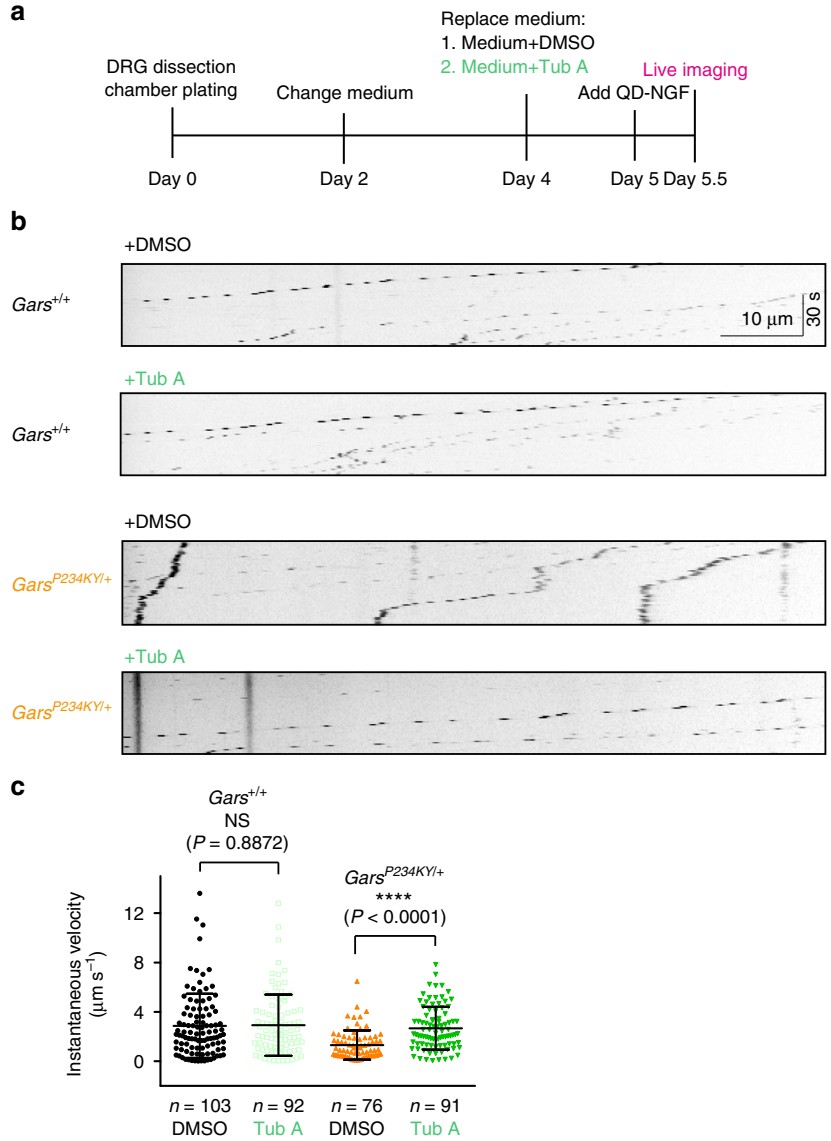

**Fig. 4** HDAC6 inhibitor restores the axonal transport defect in CMT2D mice. **a** Experimental design. **b** Representative kymographs of QD-NGF transport in *Gars*[+/+] and *Gars*[P234KY/+] DRG axons treated with either Tub A (2 μM) or solvent control (0.02% DMSO). **c** Instantaneous velocities of QD-NGF transport in DRG axons from *Gars*[+/+] and *Gars*[P234KY/+] littermates in response to Tub A treatment. DRGs were dissected and pooled from four mice for each genotype. *n* represents the number of QD-NGF-bearing endosomes measured for movements. Statistical analysis was done with two-tailed unpaired Student's *t*-test. Data are presented as means ± s.d.

**HDAC6 inhibitor improves motor functions of CMT2D mice.** The above findings prompted us to test if the HDAC6 inhibitor would be beneficial for CMT2D animals. Tub A (50 mg kg[−1] body weight) was administrated intraperitoneally to either *Gars*[+/+] or *Gars*[P234KY/+] mice starting at postnatal day 35 (P35). Again, Tub A treatment showed no significant effect on *Gars*[+/+] mice after 2 weeks of treatment (Fig. 5a–c). However, we observed a significant improvement in muscle strength based on a hindlimb extension test in the Tub A-treated *Gars*[P234KY/+] mice compared to vehicle-treated (8% captisol in saline) mice (Fig. 5a). Moreover, Tub A treatment significantly improved the motor performance of *Gars*[P234KY/+] mice in the rotarod test (Fig. 5b). Tub A-treated *Gars*[P234KY/+] animals also maintained a significantly longer walking stride compared to that of vehicle-treated group (Fig. 5c). Importantly, the behavior improvement corresponds to the significantly elevated acetylated α-tubulin levels in sciatic nerves of Tub A-treated *Gars*[P234KY/+] mice compared to vehicle-treated mice (Fig. 5d, e).

**Discussion**
We demonstrated in this study that in addition to the previously identified Nrp1 and Trk receptors[18, 20], GlyRS[CMT2D] mutant protein also aberrantly interacts with HDAC6 (Fig. 1b–d). The aberrant interaction enhances the deacetylase activity of HDAC6 and reduces the acetylation level of α-tubulin (Fig. 1e, f). The strong correlation between the intensity of the aberrant interaction and the extent of the decrease in α-tubulin acetylation level (Fig. 1g) supports the conclusion that the aberrant GlyRS[CMT2D]-HDAC6 interaction leads to HDAC6 overactivation. Remarkably, in CMT2D mice (*Gars*[P234KY/+]), a decrease in α-tubulin acetylation was specifically detected in sciatic nerves (Fig. 2a, b and Supplementary Fig. 4a, b),

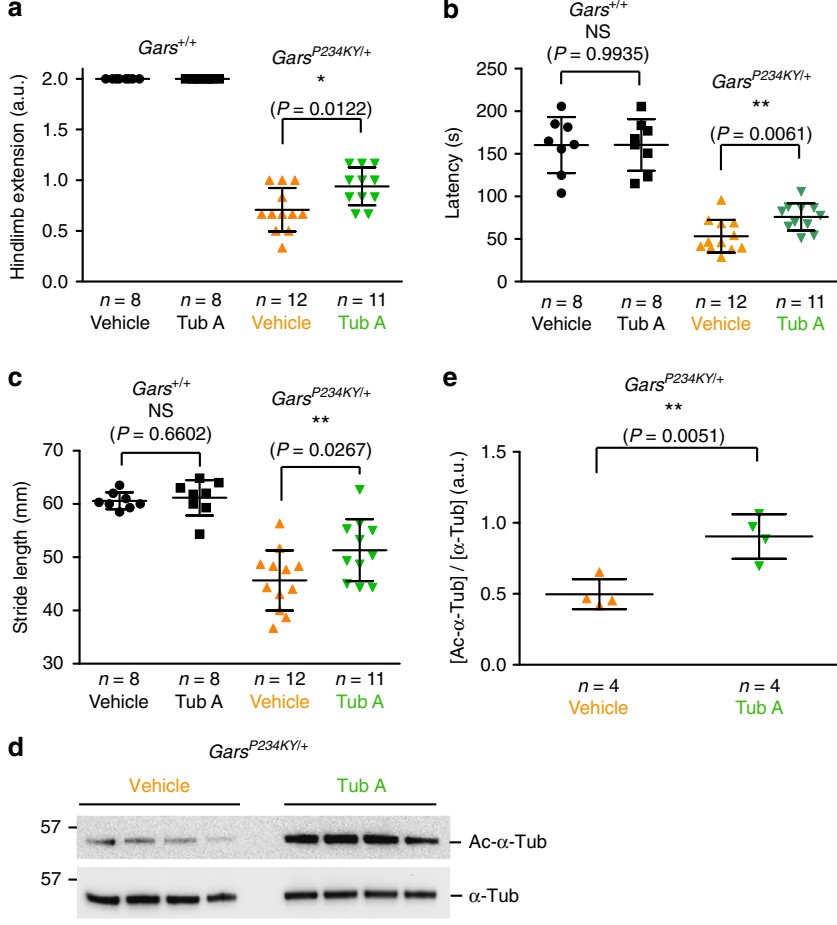

**Fig. 5** Tub A improves motor functions of CMT2D mice. **a**–**c** Hindlimb extension test (**a**), rotarod test (**b**), and walking stride measurements (**c**) on $Gars^{P234KY/+}$ and $Gars^{+/+}$ mice after 2 weeks of Tub A (50 mg kg$^{-1}$ body weight) or vehicle (8% captisol in saline) treatment starting at postnatal day 35. **d**, **e** Western blot analysis (**d**) and quantification (**e**) to evaluate the α-tubulin acetylation level in sciatic nerves of $Gars^{P234KY/+}$ mice with Tub A or vehicle treatment. The protein levels were quantified with ImageJ. Statistical analysis was done with two-tailed unpaired Student's $t$-test. $n$ represents number of mice per group. Data are presented as means ± s.d.

consistent with the peripheral nerve selectivity of the disease. In line with the established role of α-tubulin acetylation in facilitating axonal transport[26], deficits in axonal transport were found in DRG cultures derived from CMT2D mice (Fig. 3c–e and Supplementary Movies 1 and 2). It is important to stress that the reduction in α-tubulin acetylation and the axonal transport deficits were detected prior to the onset of CMT2D symptoms, suggesting that they are not consequences of axonal degeneration, but rather contribute to the cause of the disease. Therefore, the length-dependent axonal degeneration in CMT2D may be explained by axonal transport deficits caused by the aberrant GlyRS$^{CMT2D}$-HDAC6 interaction and the subsequent HDAC6 overactivation (Fig. 6). Moreover, by showing that an HDAC6 inhibitor is able to restore high levels of α-tubulin acetylation in sciatic nerves, rescue axonal transport defects, and improve motor function in CMT2D mice (Figs. 4 and 5), we demonstrate that HDAC6 is a promising therapeutic target for CMT2D.

Although the GlyRS$^{P234KY}$ mutant is ubiquitously expressed in the CMT2D mice, decrease in α-tubulin acetylation was not detected in spinal cord and brain samples (Fig. 2a, b). Interestingly, we found that in normal mice ($Gars^{+/+}$), the level of acetylated α-tubulin in sciatic nerves is substantially higher than that in spinal cord and brain (Fig. 2a, b), possibly because proportionally more axons are contained in the sciatic nerve sample, and high levels of acetylated α-tubulin may be required for

specific functions carried out within the nerve processes, such as axonal transport. We also found that the level of HDAC6 protein is lower in sciatic nerves compared with spinal cord and brain (Fig. 2a, c), which provides a plausible biochemical explanation for the above observation. Most importantly, GlyRS, in contrast to HDAC6, has a more uniform distribution, which creates a higher concentration ratio of GlyRS to HDAC6 in sciatic nerves (Fig. 2a, c), which may allow the manifestation of the gain-of-function effect of GlyRS$^{P234KY}$ on HDAC6. Therefore, we suggest that the peripheral nerve specificity of CMT2D is linked to the need for peripheral nerves to maintain high levels of acetylated α-tubulin, creating a selective sensitivity of peripheral nerves to HDAC6 overactivation.

Interestingly, unlike α-tubulin, the acetylation levels of two other HDAC6 substrates (i.e., cortactin and HSP90) are not reduced in sciatic nerves of the CMT2D mice. Also, noticeably, in normal mice ($Gars^{+/+}$), the acetylation levels of cortactin and HSP90 in sciatic nerves are not higher than that in spinal cord and brain (Fig. 2a). Possibly, consistent with the above idea, the relatively low levels of acetylated cortactin and HSP90 substrates make them less sensitive to HDAC6 activity change. It is also possible that an additional deacetylase, not affected by GlyRS$^{P234KY}$, is involved. Regardless, these observations indicate apparent selectivity in the effect of a GlyRS mutation at both the tissue level and the substrate level.

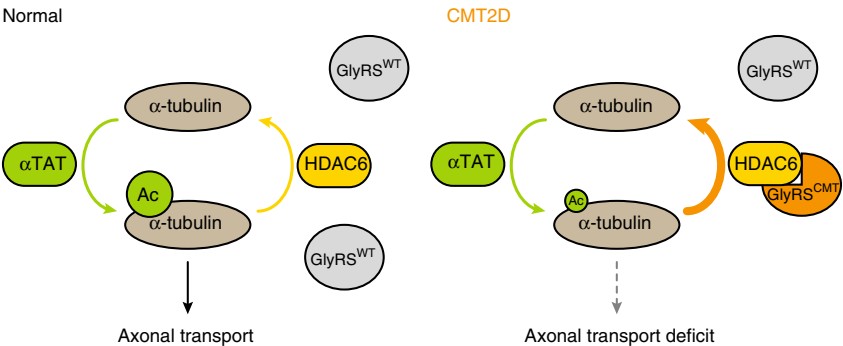

**Fig, 6** A model for the mechanism by which GlyRS$^{CMT2D}$ impairs axonal transport. Left, α-tubulin acetyl-transferase (αTAT) and HDAC6 deacetylase are two major enzymes regulating α-tubulin acetylation, which is critical for axonal transport in peripheral nerves. Right, CMT2D mutations alter the conformation of GlyRS, enabling GlyRS$^{CMT2D}$ to bind to HDAC6. This interaction aberrantly enhances the HDAC6 activity, decreases the level of α-tubulin acetylation, leading to axonal transport deficit

All GlyRS$^{CMT2D}$ mutants we tested, including the ambiguous GlyRS$^{S581L}$, induce a gain-of-function interaction with HDAC6 (Fig. 1c). However, the strength of the aberrant interaction varies substantially. Particularly, among the human mutations, S581L and G598A induce the strongest HDAC6 interaction; whereas, E71G, L129P, S211F, G240R, E279D, H418R, and G526R induce the aberrant interaction more weakly (Fig. 1c and Supplementary Fig. 3). The strong aberrant GlyRS$^{S581L}$-HDAC6 interaction may explain why the mutation is recurrently identified in CMT patients[35, 37]. Indeed, an aberrant GlyRS-HDAC6 interaction was detected in a CMT patient carrying the S581L mutation (Fig. 1d). Although many CMT2D mutations induce a relatively weak interaction with HDAC6 in NSC-34 cells, there is a clear distinction between the mutations found in CMT2D patients versus the benign variants identified in the general population, which induce no HDAC6 interaction at all (Supplementary Fig. 1). Nevertheless, the relatively weak HDAC6 interaction for many GlyRS$^{CMT2D}$ mutants suggests the possibility of having other mechanisms that influence the length-dependent vulnerability of axons in CMT2D.

It is worth noting that the reported clinical presentations of patients carrying the S581L and G598A mutations are atypical for CMT2D[35]. The S581L and G598A patients have more severe distal weakness and wasting in the lower limbs[35], in contrast to the upper limb predominance found in other CMT2D patients[6, 46]. Thus, the aberrant GlyRS$^{CMT2D}$-HDAC6 interaction appears to correlate with the divergent clinical presentation. Moreover, the G598A mutation can induce strong aberrant interactions of GlyRS with both Nrp1 and HDAC6, potentially explaining the severe, early-onset clinical symptoms of patients carrying this mutation.

Like G598A, the P234KY mutation found in mice also has the ability to induce strong aberrant interactions with both Nrp1[18] and HDAC6 (Fig. 1b, c), correlating with the severe phenotypes of this CMT2D mouse model. In previous work[18], we have demonstrated that targeting the Nrp1 signaling pathway by overexpressing vascular endothelial growth factor improves motor functions in the *Gars$^{P234KY/+}$* mice. Here, in the same mouse model, we also demonstrate that targeting HDAC6 can reach a similar level of functional improvement. Neither treatment alone has reached full recovery, suggesting potential benefit of a combination therapy that targets both pathways.

Taken together, we propose that CMT2D neuropathy results from multifactorial pathogenic mechanisms, and different disease-associated mutations may have different predominance in mechanisms that contribute to the overall pathophysiology. The

most effective therapy may need to target multiple pathways, or at least the predominantly dysregulated pathway in each patient.

## Methods

**Cell culture and western blot analysis**. HEK293 cells and NSC-34 cells were obtained from American Type Culture Collection without further authentication. Cells were cultured in Dulbecco's modified Eagle's media (11995; Gibco) supplemented with Pen/Strep (15140, Gibco) and 10% fetal bovine serum (FB-12, Omega). Sequences of human GlyRS (WT or CMT mutants) were inserted into the pcDNA6v5c plasmid for overexpression purpose. Sequences of human HDAC6 (full length or fragments) were inserted into either the pFlagCMV or the pmCherry plasmid for overexpression purpose. All transfections were done with Lipofectamine 2000 (Invitrogen) when cells reach ~80% confluence. Forty-eight hours after transfection, cells were washed with phosphate-buffered saline (PBS) and then lysed with the lysis buffer (#9803; Cell Signaling Technology; 20 mM Tris-HCl [pH 7.5], 150 mM NaCl, 1 mM Na$_2$EDTA, 1 mM EGTA, 1% Triton, 2.5 mM sodium pyrophosphate, 1 mM β-glycerophosphate, 1 mM Na$_3$VO$_4$, and 1 μg mL$^{-1}$ leupeptin) supplemented with protease inhibitor (Roche). The supernatant of the lysate was subjected to western blotting. Western blot images have been cropped for presentation. Full-size images are presented in Supplementary Fig. 6 with the information on antibody dilutions included. The antibodies used in this study include anti-GlyRS (44E9 from aTyr Pharma [proprietary], 4.4 mg mL$^{-1}$, 1:2000; sc-98614 from Santa Cruz, 0.2 mg mL$^{-1}$, 1:500), anti-V5 (R96-CUS, Invitrogen, 1.2 mg mL$^{-1}$, 1:3000), anti mCherry (5993–100, BioVision, 0.5 mg mL$^{-1}$, 1:500), anti-acetylated-cortactin (09–881, EMD Millipore, 1.0 mg mL$^{-1}$, 1:500), and anti-acetylated-HSP90 (600-401-981, Rockland immunochemical Inc., 1.0 mg mL$^{-1}$, 1:500). Other antibodies, including anti-HDAC6 (#7612, 1:1000), anti-Flag (#2908, 1:1000), anti-acetylated-α-tubulin (#5335, 1:1000), anti-α-tubulin (#3873, 1:2000), anti-cortactin (#3503, 1:1000), and anti-HSP90 (#4877, 1:1000) are from Cell Signaling. Antibody validation information is available in the product's manual.

**Immunoprecipitation**. A unit of 2 μg of anti-V5 (R96-CUS; Invitrogen), anti-HDAC6 (#7612; Cell Signaling), anti-Flag (#2368; Cell Signaling) antibodies, mouse IgG (#5415; Cell Signaling), or rabbit IgG (#3900; Cell Signaling) were coupled to 30 μl of protein G-sepharose (Amersham Biosciences) beads and used for immunoprecipitations. Supernatant of HEK293 cell lysates, NSC-34 cell lysates, or mouse brain tissue lysates were then added and incubated with the antibodies for 3 h or overnight at 4 °C. The G-sepharose beads were then washed four times with 1 mL of cold PBS buffer (pH 7.4). The bead-bound proteins were eluted and denatured with SDS loading buffer and subjected to SDS-polyacrylamide gel electrophoresis (SDS-PAGE) and western blotting.

**PBMC isolation and immunoprecipitation**. Blood samples from a CMT2D patient and a healthy donor were requested under the Baylor College of Medicine Institutional Review Board-approved consent. PBMCs were isolated from whole blood (10 mL) by following the manufacture's protocol (Lymphoprep, Stem Cell Tech). Erythrocyte contamination was removed by suspending the cell pellet in ACK lysing buffer (12002-070; Lonza Walkersville Inc.) followed by washing twice using RPMI medium (30-2001, Gibco). The cell pellet was then lysed with the lysis buffer (#9803; Cell Signaling Technology; 20 mM Tris-HCl [pH 7.5], 150 mM NaCl, 1 mM Na$_2$EDTA, 1 mM EGTA, 1% Triton, 2.5 mM sodium pyrophosphate, 1 mM β-glycerophosphate, 1 mM Na$_3$VO$_4$, and 1 μg mL$^{-1}$ leupeptin) supplemented with protease inhibitors (Roche). Protein concentrations of the

supernatants were quantified using the Bradford method against a standard curve of bovine serum albumin. The lysates were then brought to the same concentration using the lysis buffer, and equal amounts of total protein were loaded to protein G-sepharose (Amersham Biosciences) beads previously incubated with either anti-HDAC6 antibody (#7612; Cell Signaling) or anti-Rabbit IgG (#3900; Cell Signaling) overnight at 4 °C under gentle rotation. The G-sepharose beads were then washed four times with 1 mL of cold PBS buffer (pH 7.4). The bead-bound proteins were eluted and denatured with SDS loading buffer and subjected to SDS-PAGE and western blotting.

**HDAC6 deacetylase activity assay**. Fluor-de-Lys® HDAC6 fluorometric drug discovery kit (BML-AK516-0001; Enzo Life Science) was used according to the manufacturer's instruction. HEK293 cells around 80% confluence were transfected with either empty pcDNA6v5c plasmid or pcDNA6v5c containing the GlyRS (WT or P234KY) sequences (five 10 cm dishes for each). Forty-eight hours after transfection, cells were washed with PBS and then lysed with the lysis buffer. The supernatant was then used for immunoprecipitation with protein G-sepharose beads pre-incubated with anti-HDAC6 antibodies (#7612; Cell Signaling) overnight at 4 °C. The G-sepharose beads were then washed four times with 1 mL of cold PBS buffer (pH 7.4). The bead-bound proteins were incubated with the soft elution buffer (0.2% [w/v] SDS, 0.1% [V/V] Tween-20, and 50 mM Tris-HCl, pH 8.0) for 7 min at room temperature with rotation at 1000 r.p.m. The eluted proteins were used for the activity assay. Reactions were stopped after 60 min with Fluor-de-Lys® Developer II and the fluorescence was measured with the excitation wavelength at 360 nm and the emission wavelength at 460 nm.

**Mice**. WT ($Gars^{+/+}$) and P234KY-CMT2D ($Gars^{P234KY/+}$) mice used in this study are predominantly in C57BL/6J background. All animal protocols and BSL2+ safety protocols were approved by The Scripps Research Institute Institutional Animal Care and Use Committee. Daily intraperitoneal injections were started on $Gars^{+/+}$ or $Gars^{P234KY/+}$ mice at postnatal day 35 for 2 weeks (5 days on and 2 days off). Tub A (101763-516; SELLECK Chemicals) was dissolved in vehicle (8% captisole [Cydex] in saline) and administrated to mice at 50 mg kg$^{-1}$ body weight. Analyses were performed on approximately equal numbers of male and female mice selected randomly from populations. All behavioral experiments were performed in a blind manner.

Mouse brain, spinal cord, and sciatic nerve (P7) samples were minced by razor blade and then homogenized with the lysis buffer (#9803; Cell Signaling Technology; 20 mM Tris-HCl [pH 7.5], 150 mM NaCl, 1 mM Na₂EDTA, 1 mM EGTA, 1% Triton, 2.5 mM sodium pyrophosphate, 1 mM β-glycerophosphate, 1 mM Na₃VO₄, and 1 μg mL$^{-1}$ leupeptin) supplemented with protease inhibitors (Roche) on ice and centrifuged at top speed for 10 min at 4 °C. The supernatants were then collected and the total protein concentrations were determined by Bradford assay. The samples were then subjected to western blotting. Mice sciatic nerve (P12 and adult) samples were homogenized by immersing into liquid nitrogen and pulverizing with urea lysis buffer (8 M urea, 2 M thiourea, 3% SDS, 75 mM dithiothreitol, 0.03% bromophenol blue, and 0.05 M Tris-HCl, pH 6.8). The mixtures were then centrifuged at top speed for 10 min at 4 °C and the supernatants were then subjected to western blotting.

**DRG dissection and microfluidic chamber culture**. The microfluidic chambers were cleaned, sterilized, and assembled as described previously[41]. Briefly, the chambers were hand-washed in 1% Alconox and rinsed three times with Milli-Q water for 30 min each time. After immersing the clean chambers in 70% ethanol for 2 h, the chambers were sterilized on both sides for 20 min under ultraviolet and then stored in a sterile 15 cm dish sealed with Parafilm at room temperature. The coverslips used for the culture were coated with poly-L-lysine (95036–792; Trevigen) and 50 ng mL$^{-1}$ mouse laminin (10571–318; Trevigen) before the dissection. DRG neurons from postnatal day 12 mice ($Gars^{+/+}$ and $Gars^{P234KY/+}$) were dissected and cultured in microfluidic chambers as reported[47, 48]. DRGs were dissected free proximally and distally, and kept in tubes filled with 2 mL dissection medium (Hibernate A plus B27 and GlutaMax [invitrogen]) on ice. Type 1 Collagenase powder (LS004194; Worthington) was dissolved in the dissection medium at a final concentration of 1 mg mL$^{-1}$, filtered, and activated by incubating at 37 °C for 1 h. DRGs were transferred to a 15 mL tube with 5 mL dissection medium and incubated at 37 °C for 8 min. The medium was then replaced with 2 mL Collagenase solution and the tubes were incubated in 37 °C water bath for 30 min with gentle shaking every 5 min. DRGs were then triturated gently with a 1 mL pipet for about 2 min. The tubes were left on the rack for 1 min and then the medium was transferred to a new tube. The trituration was repeated three times and the supernatant was then centrifuged at 3000 × $g$ for 4 min. Cell pellets were suspended in 200 μL dissection medium. The cell suspension was seeded in the microfluidic chamber (20 μL per well) and placed in the incubator (37 °C, 5% CO₂). Thirty minutes later, growth medium (Neurobasal plus B27, GlutaMax, Penstrap, and 100 ng mL$^{-1}$ NGF) was added to each well. Growth medium was replaced every 2 days. The neurites start to grow across the microgrooves around day 3. After 4 days, the DRG cultures were subjected to QD-NGF axonal transport assay.

For the evaluation of Tub A effect, growth medium was carefully replaced with fresh growth medium containing either Tub A (2 μM) or solvent control (0.02% DMSO) after 4 days' culture. The DRG cultures were then incubated for 24 h and subjected to QD-NGF axonal transport assay.

**QD-NGF axonal transport assay**. Mono-biotinylated NGF (mBtNGF) was produced and purified as previously described[49]. Briefly, HEK293FT cells were co-transfected by two vectors containing either preproNGF or *Escherichia coli* biotin ligase BirA. Mature mBtNGF was secreted into cell media and was further purified using Ni-NTA affinity chromatography. Both the cell body compartment and the axonal compartment were rinsed and depleted of NGF in Neurobasal media for 2 h. Quantum dot-conjugated mBtNGF (QD-NGF) was prepared by mixing 50 nM mBtNGF dimer with 50 nM streptavidin-conjugated QD655 (Cat#Q10121MP; Invitrogen) in Neurobasal media on ice for 1 h. QD-NGF at a final concentration of 0.2–0.5 nM was added to the axonal chamber for 3 h. To minimize the diffusion of QD-NGF into the cell body compartment, media in the cell body compartment was maintained at a higher level than that in the axon compartment during incubation. Unbound QD-NGF was rinsed off before imaging. Live-cell imaging of axonal transport of QD-NGF signals within the proximal segments of axons was carried out using a Leica inverted microscope with a ×100 oil objective lens. The scope was equipped with an environmental chamber that was maintained at a constant temperature (37 °C) and CO₂ (5%) during live imaging. QD655 signal was visualized using a set of Texas red excitation/emission cubes. Time-lapse images were acquired at the speed of 1 frame per s for a total of 2 min and were captured using a charge-coupled devise camera. Image acquisition was carried out blindly without knowing the genotypes or treatments of the DRG cultures. Kymographs were generated from the time-lapse image series using ImageJ using a previously written up macro[47] (included in Supplementary Software 1). Transport parameters (instantaneous velocity and pause duration) were analyzed as described previously[47, 48]. Briefly, instantaneous velocity was obtained by measuring the fastest moving part of a clear trajectory of each non-stationary QD-NGF-bearing endosome. The distance and angle obtained were then used to calculate the velocity according to the imaging setting. Pause duration was obtained in a similar manner by measuring the non-moving part of a clear trajectory of each non-stationary QD-NGF-bearing endosome. The distance obtained was then used to calculate the pause duration according to the imaging setting. Statistical analysis of the datasets was performed using Graphpad Prism 5.1 with two-tailed unpaired Student's *t*-test.

**Hindlimb extension assay**. The extent of hindlimb extension of mice was observed by suspending mice via the tail. A score of 2.0 corresponds to a normal extension reflex in hindlimbs with splaying of toes. A score of 1 corresponds to clench of hindlimbs to the body with partial splaying of toes. A score of 0 corresponds to clasping hindlimbs with curled toes. A score of 1.5 or 0.5 corresponded to behaviors between 2 and 1, or between 1 and 0, respectively. Three sequential tests were performed with 5 s intervals for each mouse.

**Rotarod test**. To acclimate them to the apparatus, the mice were initially placed on the stationary rod (0 r.p.m.). This was followed by a training session with a rotation speed at 1 r.p.m. for 3 min or until a fall occurred. For testing, the rotation of the rotarod was accelerated from 0 r.p.m. with an accelerating rate (0.1 r.p.m. per s). The latency of each mouse to fall was monitored for three consecutive trials and the intra-trial interval for each animal was about 15 min. The average time of three trials was calculated and used to measure motor performance.

**Footprint test**. Blue ink was applied to the hind paws of each mouse, which was then placed at the entrance of a narrow channel (10 cm × 80 cm × 25 cm) with the floor covered with white paper. The top of the channel was covered by tissue paper and the home cage of the mice was placed at the end of the channel to attract the mice to walk through the channel while leaving its footprints on the paper. Stride length was assessed by measuring the average distances of at least three consecutive steps.

**Statistics**. All graphs and data generated in this study were analyzed using GraphPad Prism 5.1 Software (MacKiev), Origin (OriginLab), or Excel (Microsoft). We did not use statistical methods to predetermine sample sizes, but our sample sizes are comparable to those generally found in the field. Two-tailed unpaired Student's *t*-tests were used to measure differences from at least three independent replicates and no data points were excluded from the analyses for any reason. The normality of the data was determined by D'Agostino–Pearson omnibus test and Shapiro–Wilk normality test, and the *F* test has been performed to compare the variances of the data from the groups that are being statistically compared.

**Data availability**. All relevant data are available from the corresponding author upon reasonable request.

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

## Acknowledgements

We thank the CMT patient who participated in this study and aTyr Pharma for providing their proprietary monoclonal antibody for GlyRS detection. This work was supported by NIH grants R01 GM088278 (to X.-L.Y.) and R01 NS054154 (to R.W.B.), and grants from the Tau Consortium and The Larry L. Hillblom Foundation (to C.W.), and by aTyr Pharma through an agreement with The Scripps Research Institute.

## Author contributions

Z.M. and X.-L.Y. designed the study, analyzed the data, and prepared the manuscript. Z.M. performed cell culture, molecular cloning, immunoprecipitation, and other biochemical experiments. X.Z. and X.-Q.C. designed and performed the QD-NGF transport assay. Z.M., Q.H., H.L., G.B., and J.Z. carried out mice dissection. Z.M., H.L., G.B., and J.P. performed mice injections and behavior testing. N.W. assisted with GlyRS^CMT2D constructs and PBMC isolation. Z.L. assisted with mice studies. C.T.C. provided blood sample from CMT2D patient. R.W.B. provided mice, technical and

scientific advice, and assistance with the manuscript. G.B., C.W., and S.L.P. provided technical support and scientific advice.

## Additional information

**Competing interests:** X.-L.Y. is a scientific co-founder and consultant of aTyr Pharma. The remaining authors declare no competing interests.

