## [Peer Review File · Nature Communications]

Reviewers' comments:

Reviewer #1 (Remarks to the Author):

Here the authors identify a new putative molecular partner for mutant GlyRS proteins linked to the axonal degeneration syndrome CMT2D, namely HDAC6. Interaction of mutant GlyRS with HDAC6 can reduce tubulin acetylation in some, but not all cases, at least in vitro. Importantly, HDAC6 inhibition can provide some phenotypic rescue in a mouse model of CMT2D (not a human mutation). There are some conceptually appealing aspects of the work, which are lucidly laid out by the authors: linking altered tubulin regulation to an axonopathy and dissecting the combination of abnormal protein interactions leading to the full disease phenotype in CMT2D. However, the manuscript as it currently stands lacks key elements needed for appeal to a broad audience. Given that many disease linked mutant forms of GlyRS do not appear to significantly alter tubulin acetylation, it seems quite possible that another HDAC6 substrate is more important in CMT2D. Additional work would be required to provide a strong link to disease pathogenesis, particularly given that the mouse mutant studied is not one found in patients.

Specific points:

1. Editing for proper English usage is needed.
2. Fig. 1b,c. The specific coprecipitation of mutant GlyRS with HDAC6 is at first glance impressive; however, there appears to be significant binding of wild type GlyRS with HDAC6 in vivo, which is not seen in vitro, suggesting that the in vitro experiments are not accurately reflecting the in vivo situation. Further, some, but not all, disease associated GlyRS mutants reduce tubulin acetylation, which is most simply interpreted as dissociation of the tubulin acetylation effects and disease.
3. Fig. 1g. It is not entirely clear how the values were derived. If quantitative analysis of tubulin acetylation was performed, those data should be presented along with appropriate statistical analysis. If quantitative analysis was not performed, the data presentation appears to be an over interpretation.
4. Figs. 3 and 4. The phenotypic and physiological rescue seems rather modest. The authors should discuss the implications of this for any potential treatment in patients.

Reviewer #2 (Remarks to the Author):

In this paper, the authors describe the aberrant interactions of CMT2D-linked Gly-RS mutants with HDAC6. They observe an increase in HDAC6 activity in the presence of the

mutant Gly-RS, which in turn leads to a decrease in acetylated tubulin. The authors hypothesized that as a result of the mutation, the GlyRS protein interacts aberrantly with other molecules. Based on databases for potential interaction partners of GlyRS, the authors focused on HDAC6, because of its role in tubulin deacetylation. They found that GlyRS interacts with HDAC6 and that this interaction is enhanced by the mutations. They suggest that the interaction with mutant GlyRS increases HDAC6 activity resulting in decreased tubulin acetylation. They then did an axonal transport study to show that there is an effect on axonal transport and intriguingly, tubustatin A, a HDAC6 inhibitor improved motor function in a mutant mouse model.

The model is certainly interesting, but I think that direct evidence is missing. The idea that tubulin acetylation is important in axonal transport arises from the paper by Reed et al (2006) that tubulin acetylation promotes the binding of kinesin-1. However, a later paper from the same laboratory (Hammond et al., 2010) stated that "although microtubule acetylation enhances the motility of kinesin 1, the preferential translocation of kinesin 1 on axonal microtubules in polarized neuronal cells is not determined by acetylation alone, but is probably specified by a combination of tubulin modifications." There is still a leap to accept that the enhanced HDAC6 activity on alpha-tubulin is the sole cause of the defect in CMT2D. Furthermore, the axonal transport studies that are presented in this paper show a defect in retrograde transport, but I do not know if there is evidence that tubulin acetylation has an effect on the retrograde motors.

A few additional questions/comments:

1. Fig. 1c appears to contradict Fig. 1b, since there is essentially no WT signal for GlyR5-V5, whereas there is a readily detectable amount co-immunoprecipitated in 1b.
2. With regard to this point, it would be useful to show quantification of the ratios GlyRS to HDAC6 in both 1b and 1c.
3. Where would wild-type be on the plot of fig. 1g?
4. To explain the specificity of the disease to peripheral nerves, the authors suggest that this is because there is less HDAC6 in sciatic nerve, which results in more acetylated alpha-tubulin and that the increased activity of the mutant results in less acetylated alpha-tubulin and therefore transport inhibition. This argument is consistent with the hypothesis, but there is no real evidence to back it up. Is there data that shows a dependence of acetylated alpha-tubulin on the levels of HDAC6?
5. As mentioned above, the axonal transport experiments mostly show retrograde transport changes, but is there evidence that retrograde motors are affected by acetylation.
6. The TubA rescue experiments are interesting, but it would be useful to show the results of a wild-type animal to determine how much of a rescue there is and how TubA might affect a wild-type animal

Reviewer #3 (Remarks to the Author):

In an attempt to try to explain the mechanism of length-dependent features of CMT

associated with GARS mutations, Mo and colleagues looked for possible associations between GlyRS and proteins involved in axon transport. They focused on the hypothesis that transport is likely to be affected (as a means of trying to explain the length dependent features of the neuropathy) when examining a public dataset of interactions that had been identified in an affinity capture-mass spec project. They identified an association between HDAC6 and GlyRS. They show that many disease-associated variants strengthen the association between mutant GlyRS and HDAC6 and demonstrate that the association is with both HDAC6 catalytic domains and not dependent on deacetylase activity. They show that the association between HDAC6 and mutant GlyRS enhances deacetylase activity and show that the strength of the association between HDAC6 and GlyRS seem to correlate with lower levels of acetylated tubulin. In P234KY mice, this difference appears to only impact levels of acetylated tubulin in the sciatic nerve, sparing brain and spinal cord (where acetylated tubulin is present at much lower levels in wild-type mice). They then show that transport is slowed in axons of DRG primary cultures from P234KY mice and that this defect is corrected with an inhibitor of HDAC6, Tubastatin A. Finally, they show that TubA treatment results in improvement of behavioral manifestations of the disease.

The findings presented are novel and will be of interest to the field. However, the handling of CMT2D in the introduction and discussion, and specifically the discussion of the mechanisms that have been proposed, is oversimplified in a way that overstates the novelty and impact of the work that is presented. In my opinion, significant further work would be needed to clarify the specificity and impact of the HDAC6-GlyRS interaction to warrant publication in Nature Communications.

Major comments/concerns:

1. There is increasing evidence that the S581L (or S635L if the mitochondrial leader sequence is included) is a non-pathogenic variant. It is present in 16 exomes in ExAC, which is too high for a pathogenic dominant variant to be found in the healthy population for such an extraordinarily rare disease. It is important to weigh the amount of genetic evidence to support the pathogenesis of these variants if they are to be the subject of close functional analysis.

E71G (E125G) -- strong evidence of segregation in the initial Antonellis paper-- absent in ExAC

L129P (L183P) -- strong evidence of segregation in the initial Antonellis paper-- absent in ExAC

S211F (S265F) -- segregation in a small Korean pedigree with 4 individuals (3 affected, Lee et al JPNS 2012 -- absent in ExAC

G240R (G284R) -- strong evidence of segregation in the initial Antonellis paper -- absent in ExAC

E279D (E333D) -- segregation in a large Chinese pedigree (Sun et al Neurological research 2015 -- absent in ExAC

H418R (H472R) -- strong evidence of segregation in a large pedigree in Sivakumar et al -- absent in ExAC

G526R (G580R) -- good evidence in the initial Antonellis paper -- Absent in ExAC (though G580S is present in one exome)

S581L (S635L) -- genetic evidence of segregation is not presented in the initial report of this mutation (James et al. Neurology 2006 -- it was found through candidate sequencing of just GARS not an unbiased whole exome or whole genome approach) -- present in 16 exomes in ExAC

G598A (G652A) -- one patient with a de novo variant and a phenotype that would be atypically severe for GARS mutation (James et al. Neurology-- it was found through candidate sequencing of just GARS not an unbiased whole exome or whole genome approach) -- Absent in ExAC (though G652E is present)

From the above and a close reading of the associated citation, you can see that S581L/S635L has little supporting evidence for its pathogenesis and is a major outlier in the group that was examined and is likely non-pathogenic. Furthermore, this variant has arisen in additional families with CMT2 and has not segregated in at least two of these families (Griffin et al. Human Mutation 2014), also arguing that this is a non-pathogenic variant.

In light of the above, it is difficult to conclude that this HDAC6 association unifies pathogenic variants. Importantly, this does not eliminate the possibility that this association modifies the phenotype and that inhibition could be a therapeutic avenue, but the language on the finding should be more nuanced to reflect this.

Nevertheless, finding unifying mechanisms that truly differentiate disease-associated variants from non-pathogenic variants is crux to the field. To address this concern, the handling of S581L should be changed to reflect the uncertainty surrounding its pathogenesis. In light of this non-pathogenic variant having among the strongest associations with HDAC6, and lowest levels of acetylated tubulin, more work is required to better understand how this association correlates with pathogenesis. I would suggest looking at whether 4-6 additional non-pathogenic missense variants present in 10-400 exomes in ExAC and in relatively well-conserved amino acids (similar to the level of conservation in S635L) impact association with HDAC6. While clearly out of the scope of this work, effect of these rare non-pathogenic variants on open conformation and Nrp1 association would be an important addition to the field.

2. HDAC6 immunoprecipitated WT GlyRS from mice. It is surprising to me that this result could not be replicated with V5-tagged GARS in figure 1C. The authors argue that this is because the association is significantly stronger with all of the mutants that were tested. It is difficult to assess the relative strengthening of the association without some baseline for wild-type protein. This inconsistency is not addressed by the authors and the association is referred to as an aberrant interaction rather than a strengthening of a physiologic interaction.

It is surprising to me that a strengthened interaction between GlyRS and HDAC6 catalytic domain results in increased deacetylase activity. Was this counterintuitive to the authors? Is there any precedent for increased activity of HDACs when bound to a mutant protein?

3. Instantaneous velocity and average velocity need to be defined more clearly in the axon transport experiments. How was the instantaneous velocity determined and how was the

'instant' where velocity was assessed selected for analysis? Figure legends define a biological replicate as a single NGF-bearing endosome. Statistical analysis should be applied to biological replicates not technical replicates. I would suggest analyzing kymographs to break down each cargo's path into pauses and periods of motor activity and averaging the pause duration and the velocity only when the cargo is in motion. It would also be preferable to have error bars represent the standard deviation, as the high n's (based around the definition of a replicate as an individual cargo) will bring down the SEM even with very significant variability. Means and standard deviations should be listed in the figure legends throughout the paper. In summary, please clarify more clearly how this analysis was performed in your methods with references that validate the analysis and statistical approach.

The effect of TubA on transport in wild-type axons should be examined. The authors argue that this is a CMT2D specific mechanism and therefore the correction (or improvement in transport with TubA treatment) should be specific to diseased neurons. Despite this, the data presented might suggest that with TubA treatment the transport deficits improve to above the level of wild-type transport. Figure 3D shows wild-type instantaneous velocity mean ~ 2 and mutant ~ 1 . Figure 3E shows DMSO treated P234KY instantaneous velocity mean ~ 2 and TubA treated P234KY instantaneous velocity ~ 2.5 , which is above the mean for WT mice.

4. To determine whether the improvement seen with TubA treatment represents a potentially clinically meaningful change (which is especially important if this were to seriously be considered as a viable therapy, as is suggested by the paper) it is important to have age-matched wild-type controls on the same graph. This would allow the reader to evaluate whether TubA treatment approximates a return to baseline and whether it represents a clinically significant change?

5. The handling of CMT2D in this paper (including the first sentence of the abstract) implies that it is unique among neuropathies for its length dependent presentation. This is not the case as most neuropathies (genetic and acquired) present with length dependent features. The cause of the length dependence in CMT2D and in other neuropathies is probably multi-factorial and the language in the paper should be softened to reflect that there are probably several mechanisms that influence the length-dependent vulnerability of axons in CMT2D.

6. Improving the clarity and style of the writing in the body of the paper would strengthen the impact of the work. There are numerous errors in verb agreement and syntax, which should be corrected.

Minor comments:

1. figure legend for figure 1d is mislabeled with e

2. the axes of the graphs in the supplementary figures contain multiple typographical errors (sup fig 1d, protein levle)(sup fig 2b and 3c, average elocity).

3. GlyRS^{CMT2D} is not an allele, and I would consider revising the text to refer instead to mutant GlyRS or better still, disease-associated GlyRS, or disease-associated GARS variants

4. in the figure legend for supplementary video S4 misspells tubastatin A

Point-by-point Response to Reviewers' Comments:

First of all, we would like to thank all reviewers for their comments. We believe the paper has been strengthened as a result of addressing them. We have performed many additional experiments and edited the manuscript accordingly. On the revised manuscript, all major changes we made are highlighted in red; new or updated figures are indicated in green.

Reviewer #1:

Here the authors identify a new putative molecular partner for mutant GlyRS proteins linked to the axonal degeneration syndrome CMT2D, namely HDAC6. Interaction of mutant GlyRS with HDAC6 can reduce tubulin acetylation in some, but not all cases, at least in vitro. Importantly, HDAC6 inhibition can provide some phenotypic rescue in a mouse model of CMT2D (not a human mutation). There are some conceptually appealing aspects of the work, which are lucidly laid out by the authors: linking altered tubulin regulation to an axonopathy and dissecting the combination of abnormal protein interactions leading to the full disease phenotype in CMT2D. However, the manuscript as it currently stands lacks key elements needed for appeal to a broad audience. Given that many disease linked mutant forms of GlyRS do not appear to significantly alter tubulin acetylation, it seems quite possible that another HDAC6 substrate is more important in CMT2D.

In response to the suggestion, we identified two other most studied HDAC6 substrates cortactin and HSP90, both of which have been implicated in neurodegeneration. We examined the levels of the acetylated cortactin and HSP90 in CMT2D vs. WT mice. Interestingly, no difference in the levels of the acetylated cortactin and HSP90 is found in between WT and CMT2D mice in all three types of neural tissues (sciatic nerves, spinal cord, and brain). We also observed that unlike α -tubulin, the acetylation levels of cortactin and HSP90 in sciatic nerves are not high than in other types of neuronal tissues in the normal mice (*Gars*^{+/+}) (see Figure to the right). Our explanation is that the relatively low acetylation levels of cortactin and HSP90 in sciatic nerve make it less sensitive to HDAC6 activity change. It seems that there is an apparent selectivity of mutant GlyRS at both the tissue level and the substrate level.

In addition to the new data (included as **Fig. 2a** in the manuscript), we also added discussions about this point on Page 11.

Additional work would be required to provide a strong link to disease pathogenesis, particularly given that the mouse mutant studied is not one found in patients.

In order to strengthen the link of our study with the human disease, we have obtained the blood sample from a CMT patient carrying the GlyRS S581L mutation. As shown in the Figure to the

right, the aberrant GlyRS-HDAC6 interaction can be clearly detected in the patient but not in a healthy volunteer. This new data is included in the revised manuscript as **Fig. 1d**.

We would also like to point out that although the mouse mutation (P234KY) is not found in patients, there is evidence at multiple levels suggesting that the mouse mutation is not a special case and is particularly relevant to the human disease.

1) At structural level, like human mutations, P234KY induces conformational opening around the dimerization interface of GlyRS (He *et al.*, 2011; He *et al.*, 2015).

2) At molecular function level, like most human mutations, P234KY induces aberrant interactions with both Nrp1 and HDAC6 (He *et al.*, 2015 and this work).

3) At phenotypic level, the mouse model recapitulates the major hallmarks of the human neuropathy, including locomotor deficit, electrophysiological neuronal dysfunction, axonal loss and terminal degeneration (Seburn *et al.*, 2006).

Specific points:

1. Editing for proper English usage is needed.

We have extensively edited the manuscript to improve the English.

2. Fig. 1b,c. The specific coprecipitation of mutant GlyRS with HDAC6 is at first glance impressive; however, there appears to be significant binding of wild type GlyRS with HDAC6 *in vivo*, which is not seen *in vitro*, suggesting that the *in vitro* experiments are not accurately reflecting the *in vivo* situation. Further, some, but not all, disease associated GlyRS mutants reduce tubulin acetylation, which is most simply interpreted as dissociation of the tubulin acetylation effects and disease.

We recognized the apparent inconsistency in the original submission and repeated the *in vivo* co-immunoprecipitation experiment with additional IgG control that was missing from the previous experiment. It turns out that the previously observed binding of wild-type GlyRS to HDAC6 was non-specific, because it shows up with the IgG control as well (Figure to the right). We speculated that this might be due to some non-specific cross-reactivity in the Co-IP experiment, because both the anti-HDAC6 antibody used for IP and the commercial (Santa Cruz) anti-GlyRS antibody used for WB were from rabbits.

To improve the experiment, we obtained a non-commercial high quality GlyRS mouse monoclonal antibody from aTyr Pharma and used it to repeat the experiment in three biological replicates. As shown in the figure below, in all experiments, GlyRS-HDAC6 interaction was only detected in tissues of CMT2D but not in the WT mice, indicating that WT GlyRS does not bind to HDAC6 *in vivo*, consistent with our result from using the transfected cells (both HEK293 and NSC-34 cells; see below).

This new data has replaced the old data in **Fig. 1b**.

3. Fig. 1g. It is not entirely clear how the values were derived. If quantitative analysis of tubulin acetylation was performed, those data should be presented along with appropriate statistical analysis. If quantitative analysis was not performed, the data presentation appears to be an over interpretation.

We understood the concern and repeated the experiment to allow appropriate statistical analysis. We also switched from HEK293 cells to motor neuron NSC-34 cells for the new experiment. This is because motor neuron cells are most relevant to the CMT disease. Also, our Nrp1 binding study was performed in NSC-34 cells. This would allow more accurate comparison between Nrp1 and HDAC6 interactions.

As shown the figure below, binding analysis in NSC-34 cells (left) shows larger variability than in HEK293 cell (right). We think this is due to the difference in the levels of transgene expression in the two cell lines. The expression levels of GlyRS transgenes in NSC-34 cells are lower than in HEK293 cells. Possibly, a lower level of the GlyRS mutants is less likely to saturate the binding partner and therefore more likely to manifest the variability. Nevertheless, all CMT2D mutations induce the aberrant GlyRS-HDAC6 interaction, consistent with what we observed in HEK293 cells. This new data using NSC-34 cells has replaced the old data in **Fig. 1c**.

The same experiment using NSC-34 cells has been repeated 3 times to allow statistical analysis to evaluate the correlation between the strength of an aberrant GlyRS-HDAC6 interaction (Figure above, left) and the acetylation level of α -tubulin (Figure below, left).

The levels of α -tubulin, acetylated α -tubulin, and HDAC6 bound GlyRS were quantified by ImageJ and normalized against the vector control. A strong inverse correlation ($R = -0.9321$) is confirmed (Figure above, right), supporting the conclusion that aberrant GlyRS-HDAC6 interaction promotes the deacetylase activity of HDAC6 and leads to a decrease in α -tubulin acetylation.

4. Figs. 3 and 4. The phenotypic and physiological rescue seems rather modest. The authors should discuss the implications of this for any potential treatment in patients.

In response to the suggestion, we have now included age-matched wild-type mice in the experiment to indicate the level of rescue by the HDAC6 inhibitor (Tub A). As shown in the figure below, although the Tub A treatment cannot provide full rescue (left), the functional improvement is significant and is comparable to the level of rescue provided by overexpressing VEGF to target the Nrp1 pathway (right) (He *et al.*, 2015). This suggests the potential benefit of a combination therapy that targets both pathways. We add discussions on this point in the manuscript on Page 13.

Reviewer #2 (Remarks to the Author):

In this paper, the authors describe the aberrant interactions of CMT2D-linked Gly-RS mutants

with HDAC6. They observe an increase in HDAC6 activity in the presence of the mutant Gly-RS, which in turn leads to a decrease in acetylated tubulin. The authors hypothesized that as a result of the mutation, the GlyRS protein interacts aberrantly with other molecules. Based on databases for potential interaction partners of GlyRS, the authors focused on HDAC6, because of its role in tubulin deacetylation. They found that GlyRS interacts with HDAC6 and that this interaction is enhanced by the mutations. They suggest that the interaction with mutant GlyRS increases HDAC6 activity resulting in decreased tubulin acetylation. They then did an axonal transport study to show that there is an effect on axonal transport and intriguingly, tubustatin A, a HDAC6 inhibitor improved motor function in a mutant mouse model.

The model is certainly interesting, but I think that direct evidence is missing. The idea that tubulin acetylation is important in axonal transport arises from the paper by Reed et al (2006) that tubulin acetylation promotes the binding of kinesin-1. However, a later paper from the same laboratory (Hammond et al., 2010) stated that “although microtubule acetylation enhances the motility of kinesin 1, the preferential translocation of kinesin 1 on axonal microtubules in polarized neuronal cells is not determined by acetylation alone, but is probably specified by a combination of tubulin modifications.” There is still a leap to accept that the enhanced HDAC6 activity on alpha-tubulin is the sole cause of the defect in CMT2D.

We understand that translocation of kinesin 1 on axonal microtubules may not be determined by acetylation alone but rather by a combination of tubulin modifications. However, tubulin acetylation not only affects kinesin 1 but also other motor proteins and organelles as evidenced by several previous reports. For example, a previous study demonstrated that enhanced microtubule acetylation leads to the recruitment of both kinesin-1 (anterograde motor) and dynein (retrograde motor) to microtubules and thereby stimulating both anterograde and retrograde transport (Dompierre *et al.*, 2007). Consistently, another study found that microtubule acetylation increased motility of axonemal dynein (Alper *et al.*, 2014). From a disease point of view, aberrant microtubule acetylation has been linked to Amyotrophic lateral sclerosis (ALS) (De Vos and Hafezparast, 2017). Furthermore, a reduced tubulin acetylation and defective mitochondrial transport has been reported previously in another Charcot-Marie-Tooth disease mice model (d'Ydewalle *et al.*, 2011).

As for the connection to HDAC6, HDAC6 inhibition has been shown to compensate for axonal transport defects in Huntington's disease (Dompierre *et al.*, 2007), reverses axonal transport defects in motor neurons derived from FUS-linked ALS patients (Guo *et al.*, 2017), restore axonal transport and locomotor defects in a drosophila model of mutant LRRK2-associated Parkinson's disease (Godena *et al.*, 2014), and reverse mitochondrial axonal transport in mutant HSPB1-induced CMT mice model (d'Ydewalle *et al.*, 2011). Therefore, although the enhanced HDAC6 activity on alpha-tubulin might not be the sole cause of the defect in CMT2D, our study provides ample evidence to suggest that it is likely to be involved. In response to the reviewer's comment, we have revised the Discussion of the manuscript to further acknowledge the possibility of having multiple mechanisms that cause the defect in CMT2D (on Page 12).

Furthermore, the axonal transport studies that are presented in this paper show a defect in retrograde transport, but I do not know if there is evidence that tubulin acetylation has an effect on the retrograde motors.

As mentioned above, a previous study demonstrated that enhanced microtubule acetylation leads to the recruitment of both kinesin-1 (anterograde motor) and dynein (retrograde motor) to

microtubules and thereby stimulating both anterograde and retrograde transport (Dompierre *et al.*, 2007).

A few additional questions/comments:

1. Fig. 1c appears to contradict Fig. 1b, since there is essentially no WT signal for GlyR5-V5, whereas there is a readily detectable amount co-immunoprecipitated in 1b.

Reviewer 1 also raised the same concern (specific point #2). For convenience, the response is copied below:

We recognized the apparent inconsistency in the original submission and repeated the *in vivo* co-immunoprecipitation experiment with additional IgG control that was missing from the previous experiment. It turns out that the previously observed binding of wild-type GlyRS to HDAC6 was non-specific, because it shows up with the IgG control as well (Figure to the right). We speculated that this might be due to some non-specific cross-reactivity in the Co-IP experiment, because both the anti-HDAC6 antibody used for IP and the commercial (Santa Cruz) anti-GlyRS antibody used for WB were from rabbits.

To improve the experiment, we obtained a non-commercial high quality GlyRS mouse monoclonal antibody from aTyr Pharma and used it to repeat the experiment in three biological

replicates. As shown in the figure above, in all experiments, GlyRS-HDAC6 interaction was only detected in tissues of CMT2D but not in the WT mice, indicating that WT GlyRS does not bind to HDAC6 *in vivo*, consistent with our result from using the transfected cells (both HEK293 and NSC-34 cells; see below).

This new data has replaced the old data in **Fig. 1b**.

2. With regard to this point, it would be useful to show quantification of the ratios GlyRS to HDAC6 in both 1b and 1c.

Because of the extremely clean results we now have for **Fig. 1b** in three biological replicates (see above), we felt that quantification may not be necessary any more for this experiment. If the reviewer feels otherwise, we can easily add the quantification analysis.

As for **Fig. 1c**, we have repeated the experiment using motor neuron NSC-34 cells. We switched from HEK293 cells to motor neuron NSC-34 cells for the new experiment, because motor neuron cells are most relevant to the CMT disease. Also, our Nrp1 binding study was

performed in NSC-34 cells. This would allow more accurate comparison between Nrp1 and HDAC6 interactions.

As shown the figure below, binding analysis in NSC-34 cells (left) shows larger variability than in HEK293 cell (right). We think this is due to the difference in the levels of transgene expression in the two cell lines. The expression levels of GlyRS transgenes in NSC-34 cells are lower than in HEK293 cells. Possibly, a lower level of the GlyRS mutants is less likely to saturate the binding partner and therefore more likely to manifest the variability. Nevertheless, all CMT2D mutations induce the aberrant GlyRS-HDAC6 interaction, consistent with what we observed in HEK293 cells.

Quantification of the NSC-34 cell based Co-IP experiment is shown in the figure below and added to the manuscript as **supplementary Fig. 3**.

3. Where would wild-type be on the plot of fig. 1g?

The wild-type GlyRS is now included in the new **Fig. 1g**

4. To explain the specificity of the disease to peripheral nerves, the authors suggest that this is because there is less HDAC6 in sciatic nerve, which results in more acetylated alpha-tubulin and that the increased activity of the mutant results in less acetylated alpha-tubulin and therefore transport inhibition. This argument is consistent with the hypothesis, but there is no real evidence to back it up. Is there data that shows a dependence of acetylated alpha-tubulin on the levels of HDAC6?

There is plenty evidence showing a dependence of acetylated alpha-tubulin on the levels of HDAC6. It was first reported by Hubbert *et al.* and later confirmed by several groups (Hubbert *et al.*, 2002; Matsuyama *et al.*, 2002; Zhang *et al.*, 2003) and our own data showing here. In the figure to the left, we show here our own data confirming that overexpression of HDAC6 leads to a significant decrease of acetylated alpha-tubulin in HEK293 cells.

5. As mentioned above, the axonal transport experiments mostly show retrograde transport changes, but is there evidence that retrograde motors are affected by acetylation.

As stated above, there is evidence to show that tubulin acetylation has an effect on the retrograde motors. For example, previous study demonstrated that enhanced microtubule acetylation leads to the recruitment of both kinesin-1 (anterograde motor) and dynein (retrograde motor) to microtubules and thereby stimulating both anterograde and retrograde transport (Dompierre *et al.*, 2007). Another study also found that microtubule acetylation increased motility of axonemal dynein (Alper *et al.*, 2014).

6. The TubA rescue experiments are interesting, but it would be useful to show the results of a wild-type animal to determine how much of a rescue there is and how TubA might affect a wild-type animal

In response to the suggestion, we have now included age-matched wild-type mice in all animal experiment to indicate the level of rescue by the HDAC6 inhibitor (Tub A). Please see the new **Fig. 4 and Fig. 5** in the manuscript for data.

Although the TubA treatment only provides partial rescue, the functional improvement is significant and is comparable to the level of rescue provided by overexpressing VEGF to target the Nrp1 pathway (He *et al.*, 2015). See Figure below for comparison. This suggests the potential benefit of a combination therapy that targets both pathways. We add discussions on this point in the manuscript on Page 13.

Reviewer #3 (Remarks to the Author):

In an attempt to try to explain the mechanism of length-dependent features of CMT associated with GARS mutations, Mo and colleagues looked for possible associations between GlyRS and proteins involved in axon transport. They focused on the hypothesis that transport is likely to be affected (as a means of trying to explain the length dependent features of the neuropathy) when examining a public dataset of interactions that had been identified in an affinity capture-mass spec project. They identified an association between HDAC6 and GlyRS. They show that many disease-associated variants strengthen the association between mutant GlyRS and HDAC6 and demonstrate that the association is with both HDAC6 catalytic domains and not dependent on deacetylase activity. They show that the association between HDAC6 and mutant GlyRS enhances deacetylase activity and show that the strength of the association between HDAC6 and GlyRS seem to correlate with lower levels of acetylated tubulin. In P234KY mice, this difference appears to only impact levels of acetylated tubulin in the sciatic nerve, sparing brain and spinal cord (where acetylated tubulin is present at much lower levels in wild-type mice). They then show that transport is slowed in axons of DRG primary cultures from P234KY mice and that this defect is corrected with an inhibitor of HDAC6, Tubastatin A. Finally, they show that TubA treatment results in improvement of behavioral manifestations of the disease.

The findings presented are novel and will be of interest to the field. However, the handling of CMT2D in the introduction and discussion, and specifically the discussion of the mechanisms that have been proposed, is oversimplified in a way that overstates the novelty and impact of the work that is presented. In my opinion, significant further work would be needed to clarify the specificity and impact of the HDAC6-GlyRS interaction to warrant publication in Nature Communications.

Major comments/concerns:

1. There is increasing evidence that the S581L (or S635L if the mitochondrial leader sequence is included) is a non-pathogenic variant. It is present in 16 exomes in ExAC, which is too high for a pathogenic dominant variant to be found in the healthy population for such an extraordinarily rare disease. It is important to weigh the amount of genetic evidence to support the pathogenesis of these variants if they are to be the subject of close functional analysis.

E71G (E125G) -- strong evidence of segregation in the initial Antonellis paper-- absent in ExAC
L129P (L183P) -- strong evidence of segregation in the initial Antonellis paper-- absent in ExAC
S211F (S265F) -- segregation in a small Korean pedigree with 4 individuals (3 affected, Lee et al JPNS 2012 -- absent in ExAC
G240R (G284R) -- strong evidence of segregation in the initial Antonellis paper -- absent in ExAC
E279D (E333D) -- segregation in a large Chinese pedigree (Sun et al Neurological research 2015 -- absent in ExAC
H418R (H472R) -- strong evidence of segregation in a large pedigree in Sivakumar et al -- absent in ExAC
G526R (G580R) -- good evidence in the initial Antonellis paper -- Absent in ExAC (though G580S is present in one exome)
S581L (S635L) -- genetic evidence of segregation is not presented in the initial report of this mutation (James et al. Neurology 2006 -- it was found through candidate sequencing of just GARS not an unbiased whole exome or whole genome approach) -- present in 16 exomes in ExAC
G598A (G652A) -- one patient with a de novo variant and a phenotype that would be atypically severe for GARS mutation (James et al. Neurology-- it was found through candidate sequencing of just GARS not an unbiased whole exome or whole genome approach) -- Absent in ExAC (though G652E is present)

From the above and a close reading of the associated citation, you can see that S581L/S635L has little supporting evidence for its pathogenesis and is a major outlier in the group that was examined and is likely non-pathogenic. Furthermore, this variant has arisen in additional families with CMT2 and has not segregated in at least two of these families (Griffin et al. Human Mutation 2014), also arguing that this is a non-pathogenic variant.

In light of the above, it is difficult to conclude that this HDAC6 association unifies pathogenic variants. Importantly, this does not eliminate the possibility that this association modifies the phenotype and that inhibition could be a therapeutic avenue, but the language on the finding should be more nuanced to reflect this.

Nevertheless, finding unifying mechanisms that truly differentiate disease-associated variants from non-pathogenic variants is crux to the field. To address this concern, the handling of S581L should be changed to reflect the uncertainty surrounding its pathogenesis. In light of this non-pathogenic variant having among the strongest associations with HDAC6, and lowest levels of acetylated tubulin, more work is required to better understand how this association correlates with pathogenesis. I would suggest looking at whether 4-6 additional non-pathogenic missense variants present in 10-400 exomes in ExAC and in relatively well-conserved amino acids (similar to the level of conservation in S635L) impact association with HDAC6. While clearly out of the scope of this work, effect of these rare non-pathogenic variants on open conformation and Nrp1 association would be an important addition to the field.

Thank you for pointing out the controversial on the S581L mutation. To clarify, we extensively evaluated the mutation in several ways:

1) Following the suggestion, we examined 5 missense mutations in well-conserved amino acids (R47H, V134I, A201V, T214I, M227I) identified in the general population with similar frequencies as that of the S581L mutation as reported by the Exome Aggregation Consortium (<http://exac.broadinstitute.org>). Remarkably, none of the five mutations induces an aberrant HDAC6 interaction, indicating the unique capacity of S581L among other mutations identified in the general population. This data is shown in the figure on the right and is included in the revised manuscript (**Supplementary Fig. 1**).

2) We obtained blood sample from a CMT patient carrying the S581L mutation through Prof. Dr. Thomas Caskey. A clear GlyRS-HDAC6 interaction is detected in the patient sample by not at all in the blood of a healthy donor (Figure below). This data is included in the revised manuscript as **Fig. 1d**.

3) [redacted]

Taken together, further investigations on the S581L mutation indicate that the S581L mutation is not

innocuous.

2. HDAC6 immunoprecipitated WT GlyRS from mice. It is surprising to me that this result could not be replicated with V5-tagged GARS in figure 1C. The authors argue that this is because the association is significantly stronger with all of the mutants that were tested. It is difficult to assess the relative strengthening of the association without some baseline for wild-type protein. This inconsistency is not addressed by the authors and the association is referred to as an aberrant interaction rather than a strengthening of a physiologic interaction.

Both Reviewers 1 and 2 have raised the same concern. For convenience, the response is copied below:

We recognized the apparent inconsistency in the original submission and repeated the *in vivo* co-immunoprecipitation experiment with additional IgG control that was missing from the previous experiment. It turns out that the previously observed

binding of wild-type GlyRS to HDAC6 was non-specific, because it shows up with the IgG control as well (Figure to the right). We speculated that this might be due to some non-specific cross-reactivity in the Co-IP experiment, because both the anti-HDAC6 antibody used for IP and the commercial (Santa Cruz) anti-GlyRS antibody used for WB were from rabbits.

To improve the experiment, we obtained a non-commercial high quality GlyRS mouse monoclonal antibody from aTyr Pharma and used it to repeat the experiment in three biological

replicates. As shown in the figure above, in all experiments, GlyRS-HDAC6 interaction was only detected in tissues of CMT2D but not in the WT mice, indicating that WT GlyRS does not bind to HDAC6 *in vivo*, consistent with our result from using the transfected cells (HEK293 and NSC-34 cells).

This new data has replaced the old data in **Fig. 1b**.

It is surprising to me that a strengthened interaction between GlyRS and HDAC6 catalytic domain results in increased deacetylase activity. Was this counterintuitive to the authors? Is there any precedent for increased activity of HDACs when bound to a mutant protein?

Although we are not aware of any precedent for increased activity of HDACs when bound to a mutant protein, our data suggest this is the case for CMT2D-associated mutants. The three mutations that induce strong HDAC6 interactions (i.e., P234KY, S581L, and G598A) (**Fig. 1c**) also show greatly reduced levels of acetylated α -tubulin as shown in **Fig. 1f** in the revised manuscript). And a strong inverse correlation was found between the strength of an aberrant GlyRS-HDAC6 interaction and the acetylation level of acetylated α -tubulin as shown in **Fig. 1c, f, g**). These data support the conclusion that aberrant GlyRS-HDAC6 interaction promotes the deacetylase activity of HDAC6 and leads to a decrease in of acetylated α -tubulin. These experiments have now been performed in two different cell lines (NSC-34 and HEK293) and yield consistent results supporting our conclusion.

3. Instantaneous velocity and average velocity need to be defined more clearly in the axon transport experiments. How was the instantaneous velocity determined and how was the 'instant' where velocity was assessed selected for analysis? Figure legends define a biological replicate as a single NGF-bearing endosome. Statistical analysis should be applied to biological replicates not technical replicates. I would suggest analyzing kymographs to break down each cargo's path into pauses and periods of motor activity and averaging the pause duration and the velocity only when the cargo is in motion. It would also be preferable to have error bars represent the standard deviation, as the high n's (based around the definition of a replicate as an individual cargo) will bring down the SEM even with very significant variability. Means and standard deviations should be listed in the figure legends throughout the paper. In summary, please clarify more clearly how this analysis was performed in your methods with references that validate the analysis and statistical approach.

As suggested, all plots have been converted to show the means and standard deviations throughout the revised manuscript. Detailed description on kymograph analysis was also added in the Methods section as suggested. Figure legends have also been updated for clarifications.

The effect of TubA on transport in wild-type axons should be examined. The authors argue that this is a CMT2D specific mechanism and therefore the correction (or improvement in transport with TubA treatment) should be specific to diseased neurons. Despite this, the data presented might suggest that with TubA treatment the transport deficits improve to above the level of wild-type transport. Figure 3D shows wild-type instantaneous velocity mean ~2 and mutant ~1. Figure 3E shows DMSO treated P234KY instantaneous velocity mean ~2 and TubA treated P234KY instantaneous velocity ~2.5, which is above the mean for WT mice.

There is a difference in the experiment setting for evaluating the axonal transport defects in CMT2D mice and for evaluating the efficacy of the Tubastatin A treatment. As showed below, for the initial experiment evaluating the axonal transport defects, the DRG neuron culture was maintained in the microfluidic chamber for four days after dissection to allow the axons grow across the microgrooves. Addition of QD-NGF and image acquisition was performed immediately after to compare wild type

and CMT2D mice. However, for the second experiment evaluating the efficacy of the HDAC6 inhibitor in phenotype rescue, the DRG neurons were cultured for one additional day to allow the incubation with Tubastatin A. This difference in experiment setting is likely to be responsible for the observed difference in the axonal transport velocities between these two experiments.

To solve the confusion, we have added a wild-type group (with and without Tubastatin A treatment) using the same experimental setting as for the CMT2D mice group. The result shows that 1) Tub A treatment (2μM, 24h) does not affect QD-NGF transport in the axons of wild-type (*Gars*^{+/+}) DRG; and 2) for CMT2D mice, the axonal transport velocity recovery resulting from the Tub A treatment is not above the wild-type level (figure to the left). We have updated **Fig. 4** and its description in the revised manuscript.

4. To determine whether the improvement seen with TubA treatment represents a potentially clinically meaningful change (which is especially important if this were to seriously be considered as a viable therapy, as is suggested by the paper) it is important to have age-

matched wild-type controls on the same graph. This would allow the reader to evaluate whether TubA treatment approximates a return to baseline and whether it represents a clinically significant change?

Both Reviewers 1 and 2 have raised a similar concern. In response to the suggestion, we have now included age-matched wild type mice in all animal experiment to indicate the level of rescue by the HDAC6 inhibitor (Tub A). Please see the updated **Figure 5** in the manuscript for data.

Although the TubA treatment only provides partial rescue, the functional improvement is significant and is comparable to the level of rescue provided by overexpressing VEGF to target the Nrp1 pathway (He *et al.*, 2015). See Figure below for comparison. This suggests the potential benefit of a combination therapy that targets both pathways. We add discussions on this point in the manuscript on Page 13.

5. The handling of CMT2D in this paper (including the first sentence of the abstract) implies that it is unique among neuropathies for its length dependent presentation. This is not the case as most neuropathies (genetic and acquired) present with length dependent features. The cause of the length dependence in CMT2D and in other neuropathies is probably multi-factorial and the language in the paper should be softened to reflect that there are probably several mechanisms that influence the length-dependent vulnerability of axons in CMT2D.

As suggested, we have revised the abstract to remove the confusion, and revise the manuscript to acknowledge the possibility of having several mechanisms that influence the length-dependent vulnerability of axons in CMT2D (on Page 12).

6. Improving the clarity and style of the writing in the body of the paper would strengthen the impact of the work. There are numerous errors in verb agreement and syntax, which should be corrected.

We have extensively and carefully revised the manuscript to improve clarity and style.

Minor comments:

1. figure legend for figure 1d is mislabeled with e

We have corrected it accordingly.

2. the axes of the graphs in the supplementary figures contain multiple typographical errors (sup fig 1d, protein levle) (sup fig 2b and 3c, average elocity).

We have corrected them accordingly.

3. GlyRS^{CMT2D} is not an allele, and I would consider revising the text to refer instead to mutant GlyRS or better still, disease-associated GlyRS, or disease-associated GARS variants

We use GlyRS^{CMT2D} to refer to CMT2D-associated GlyRS mutant proteins, not the gene allele. Because of the large number of CMT2D-associated mutations identified, having a general acronym helps us to improve clarity in a simple way.

4. in the figure legend for supplementary video S4 misspells tubastatin A

We have corrected it accordingly.

References:

1. Alper, J.D., Decker, F., Agana, B., Howard, J., 2014. The motility of axonemal dynein is regulated by the tubulin code. *Biophysical journal* 107, 2872-2880.
2. d'Ydewalle, C., Krishnan, J., Chiheb, D.M., Van Damme, P., Irobi, J., Kozikowski, A.P., Vanden Berghe, P., Timmerman, V., Robberecht, W., Van Den Bosch, L., 2011. HDAC6 inhibitors reverse axonal loss in a mouse model of mutant HSPB1-induced Charcot-Marie-Tooth disease. *Nature medicine* 17, 968-974.
3. De Vos, K.J., Hafezparast, M., 2017. Neurobiology of axonal transport defects in motor neuron diseases: Opportunities for translational research? *Neurobiology of disease* 105, 283-299.
4. Dompierre, J.P., Godin, J.D., Charrin, B.C., Cordelieres, F.P., King, S.J., Humbert, S., Saudou, F., 2007. Histone deacetylase 6 inhibition compensates for the transport deficit in Huntington's disease by increasing tubulin acetylation. *The Journal of neuroscience : the official journal of the Society for Neuroscience* 27, 3571-3583.
5. Godena, V.K., Brookes-Hocking, N., Moller, A., Shaw, G., Oswald, M., Sancho, R.M., Miller, C.C., Whitworth, A.J., De Vos, K.J., 2014. Increasing microtubule acetylation rescues axonal transport and locomotor deficits caused by LRRK2 Roc-COR domain mutations. *Nature communications* 5, 5245.
6. Guo, W., Naujock, M., Fumagalli, L., Vandoorne, T., Baatsen, P., Boon, R., Ordovas, L., Patel, A., Welters, M., Vanwelden, T., Geens, N., Tricot, T., Benoy, V., Steyaert, J., Lefebvre-Omar, C., Boesmans, W., Jarpe, M., Sternecker, J., Wegner, F., Petri, S., Bohl, D., Vanden Berghe, P., Robberecht, W., Van Damme, P., Verfaillie, C., Van Den Bosch, L., 2017. HDAC6 inhibition reverses axonal transport defects in motor neurons derived from FUS-ALS patients. *Nature communications* 8, 861.
7. He, W., Bai, G., Zhou, H., Wei, N., White, N.M., Lauer, J., Liu, H., Shi, Y., Dumitru, C.D., Lettieri, K., Shubayev, V., Jordanova, A., Guerguelcheva, V., Griffin, P.R., Burgess, R.W., Pfaff, S.L., Yang, X.L., 2015. CMT2D neuropathy is linked to the neomorphic binding activity of glycyl-tRNA synthetase. *Nature* 526, 710-714.
8. He, W., Zhang, H.M., Chong, Y.E., Guo, M., Marshall, A.G., Yang, X.L., 2011. Dispersed disease-causing neomorphic mutations on a single protein promote the same localized conformational opening. *Proceedings of the National Academy of Sciences of the United States of America* 108, 12307-12312.

9. Hubbert, C., Guardiola, A., Shao, R., Kawaguchi, Y., Ito, A., Nixon, A., Yoshida, M., Wang, X.F., Yao, T.P., 2002. HDAC6 is a microtubule-associated deacetylase. *Nature* 417, 455-458.
10. Matsuyama, A., Shimazu, T., Sumida, Y., Saito, A., Yoshimatsu, Y., Seigneurin-Berny, D., Osada, H., Komatsu, Y., Nishino, N., Khochbin, S., Horinouchi, S., Yoshida, M., 2002. In vivo destabilization of dynamic microtubules by HDAC6-mediated deacetylation. *The EMBO journal* 21, 6820-6831.
11. Seburn, K.L., Nangle, L.A., Cox, G.A., Schimmel, P., Burgess, R.W., 2006. An active dominant mutation of glycyl-tRNA synthetase causes neuropathy in a Charcot-Marie-Tooth 2D mouse model. *Neuron* 51, 715-726.
12. Zhang, Y., Li, N., Caron, C., Matthias, G., Hess, D., Khochbin, S., Matthias, P., 2003. HDAC-6 interacts with and deacetylates tubulin and microtubules in vivo. *The EMBO journal* 22, 1168-1179.

[redacted]

REVIEWERS' COMMENTS:

Reviewer #1 (Remarks to the Author):

The authors have improved their biochemical evidence linking mutant GlyRS and HDAC6 to tubulin acetylation and the pathogenesis of CMT2D. While some questions do remain, particularly regarding the general nature of the effect on tubulin acetylation and the role of altered axonal transport, overall these revised findings are now sufficiently compelling to be of broad interest.

Reviewer #2 (Remarks to the Author):

The authors present additional data that are more convincing

Reviewer #3 (Remarks to the Author):

Mo and colleagues performed a significant amount of work to address my concerns. I will respond to each of my concerns and how they've been addressed. Overall, while I think I still have reservations about the significance of the HDAC6 association and its clinical relevance in CMT2D, I think that this work is now worthy of publication in Nature Communications.

1: They have add 5 important benign variant controls to the western blots showing association with HDAC6. This helps argue that there is disease specificity to this association.

[redacted]

2. Use of the new antibody has improved the evidence that the association is unique to mutant protein.

3. The characterization and representation of the instantaneous velocity and average velocity measurements in the methods and results has improved. It still concerns me that they use single axon replicates rather than biological (dish or animal) replicates. I worry that there is inappropriate statistical weight with such an artificially inflated number of replicates (e.g. $n > 100$). With the error bars it is clear that confidence intervals overlap significantly. Despite this there is great statistical significance. We can let the readers decide if this is a valid difference based on the experimental design that has been more thoroughly described.

4. Inclusion of age matched WT controls improves this experiment.

5. There is a bit more nuance to the handling of neuropathy as a length dependent process. There are parts of the text where language continues to hamper the arguments being made. For example in a discussion of Figures 2a and 2b, ((Fig. 2a, b), possibly because more nerve processes and less cell bodies are contained in the sciatic nerve sample,) they seem to discount or ignore all glial cell bodies and the long tract axons present in the spinal cord.

While improved, the writing still suffers from numerous errors in verb agreement and stylistic errors that detract from the arguments being made. I would hope that these could be corrected before a final draft is published. Perhaps the editors or copy editors at Nature Communications could help polish the prose.